# Monitoring of *Dactylorhiza sambucina* (L.) Soó (Orchidaceae)—Variation in Flowering, Flower Colour Morph Frequencies, and Erratic Population Census Trends

**Matthias Kropf *** and **Monika Kriechbaum**

Institute for Integrative Nature Conservation Research, University of Natural Resources and Life Sciences, Vienna (BOKU), Gregor-Mendel-Str. 33, 1180 Vienna, Austria
* Correspondence: matthias.kropf@boku.ac.at

**Abstract:** Central questions of reproductive research on *Dactylorhiza sambucina* (Orchidaceae) are, if and how pollinator-mediated negative frequency dependent selection might maintain its flower colour polymorphism. As this hypothesis was based on artificial populations, it needs to be verified under natural conditions. Therefore, we demonstrate and discuss spatial-temporal variation of flowering, flower colour morph frequencies and population fluctuations in *D. sambucina* as prerequisites for understanding its population and reproduction dynamics. Knowledge of these dynamics is also crucial for the species' conservation. We investigated colour morph frequencies for ten Austrian populations of *D. sambucina* over maximum time series of 18 consecutive years. We report repeated changes in the dominance of red- or yellow-flowering morphs in at least three populations during these time series. Even though being basically erratic (e.g., switches in different years), we identify smaller populations as being more prone to changes of flower colour dominance. Conversely, constant morph frequencies and the dominance of one flower colour morph is demonstrated for individual-poor and large populations. As previous large-scale (continental) analyses failed to identify environmental factors explaining the distribution of differing morph frequencies across Europe, we strongly argue for local approaches by investigating such factors at the micro-habitat scale.

**Keywords:** *Dactylorhiza sambucina*; flower colour polymorphism; flowering dynamics; generalized food-deception; Lower Austria; monitoring; negative frequency-dependent selection; population census and dynamics

## 1. Introduction

Within the orchid family (Orchidaceae), representing the most species-rich monocotyledonous plant family with about 19,000 known species [1], a high proportion, i.e., about one-third up to half of all species, is characterized by a deceptive pollination strategy [2–4]. For the c. 450 European terrestrial orchids compiled data [5] indicate about 13% of the species are rewarding by nectar, c. 5.5% by offering sleeping sites, and the overwhelming portion of about 81.5% of these orchid species follow a deceptive pollination strategy. Within the latter group the majority of taxa acts by sexual swindle [genus *Ophrys*, (cf. [6])], while nearly one-third of the cheating European orchid species are food-deceptive [5]. The Elder-flowered Orchid (*Dactylorhiza sambucina* [L.] Soó) represents a European species that is effectively pollinated by bumblebees. In the early spring, typically naïve bumblebee queens are cheated, as *D. sambucina* provides no nectar reward, although the species is morphologically characterised by having an (empty) flower spur, [7–11]. This characteristic of *D. sambucina* was already observed in 1790 (named *Orchis latifolia* at that time) by the famous pollination biologist Christian Konrad Sprengel, who introduced the term "Scheinsaftblume" [12]. In this sense, *D. sambucina* represents a so-called generalized food-deceptive species [3,13,14], as different authors failed to identify natural specific model

plants that might be mimicked [8,11,15]. However, an experimental study [16] could demonstrate that facilitation by co-flowering rewarding plant species is possible in *D. sambucina*, i.e., demonstrating a magnet species effect [17,18]. Furthermore, *D. sambucina* usually is characterised by a striking flower colour dimorphism with yellow- and red-flowering individuals occurring together (Figure 1) in varying frequencies all over Europe (cf. [19]): i.e., in Sweden (red-biased; [8,20,21]); in Austria [9,22,23]; in Germany (yellow-biased; [24–26]); in France (yellow-biased; [15,27,28]); in Italy (yellow-biased; [29,30]); in the Czech Republic (red-biased; [31–33]; in Switzerland (yellow-biased; [34]). The mix of yellow- and red-flowering individuals within populations should (by a "doubled" chance attracting pollinators) promote pollinator switches between morphs, and, therefore, favour cross-pollination [8,9]. However, an experimental study found no evidence that polymorphic arrays had higher mean reproductive success compared to monomorphic [35], likewise reproductive success was not reduced in monochromic yellow-flowering *D. sambucina* populations under natural conditions [36], and a pollen tracking study [37] could not demonstrate predominant switches between colour morphs.

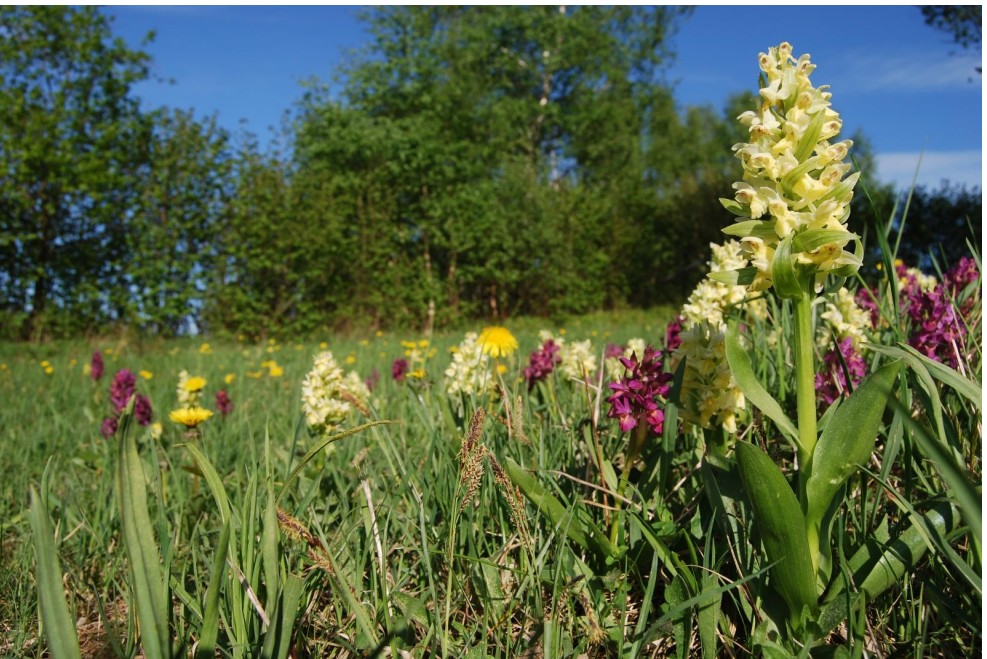

**Figure 1.** *Dactylorhiza sambucina* flowering in a Waldviertel population. The photograph shows yellow- and red-flowering individuals, which were monitored in ten different Lower Austrian populations to demonstrate natural flower morph fluctuations (photo credits: M. Kropf, 5 May 2015).

Based on field experiments using artificially arranged populations with differing morph frequencies, Gigord et al. [27] showed that negative frequency-dependent selection (NFDS) could stabilize the flower colour dimorphism in the long term. These authors demonstrated that the (extremely) rare morph is characterised by a disproportionally higher reproductive success (here: fruit set) compared to the dominating morph. This relatively higher reproductive success could then lead to the establishment of new plants representing the rare morph; thus, preventing its extinction. By doing so, different morph ratios could theoretically be stabilized; in the original experiment with *D. sambucina* the equilibrium was about 2/3 yellow- to 1/3 red-flowering plants [27].

However, other research teams could not verify the hypothesised disproportionally higher reproductive success of the, respectively, rare morph under natural conditions (e.g., in Italy [29]; in the Czech Republic [32]). The latter authors, while working in populations dominated by the red-flowering morph, rather showed by (all possible) crossing experiments that seed mass and seed viabilitywere higher in red-flowering mother plants

(i.e., pollen acceptors), what might explain the dominance of this morph [32]. In similar crossing experiments in Switzerland, Juillet et al. [34], in contrast, found marginally significant reduced survival rates of the offspring of red-flowering plants. This finding corresponds well to the dominance of the yellow-flowering morph there. Therefore, it seems likely that additional factors, and not just the choice by pollinating bumblebees, are influencing the morph frequencies of *D. sambucina* populations.

With respect to the behaviour of pollinators in a mixed (yellow-dominated) population of *D. sambucina,* the pollen tracking study by Groiß et al. [37] could not demonstrate predominant switches between morphs as mentioned above. Even after statistically correcting for the yellow-dominance in the study population, pollen from yellow flowers landed more often on yellow flowers than expected by chance; additionally, the same was true for the pollen from red flowers landing again on red flowers (Chi$^2$-test). Therefore, these findings argue more for pollinator consistency and assortative mating [37] than for preferential pollinator switches as implied by the principle of "two times cheating" by two flower colour morphs.

On the continental scale, morph frequencies are differing a lot: while in the Cevennes in France, where Gigord et al. [27] performed their NFDS experiments, the natural populations are characterised by a ratio of 2/3 yellow- to 1/3 red-flowering plants, in the Czech Republic red-flowering individuals are dominating [32]. In Germany, there are almost exclusively yellow-flowering populations (cf. [36,38,39]). Based on the reproduction success in these populations, which was not lower than in populations with both flower colour morphs, Kropf and Renner [36] argued that the absence of the second flower colour has no obvious negative impact. Consequently, other factors may influence the reproduction success and therefore dominance of yellow-flowering individuals [36].

In general, and independent from the (crossing) experiments performed, we have to ask whether (medium-term) changes in the flower colour morph frequencies are observable in natural populations. Therefore, we also need to know, for instance, how population dynamics, in terms of varying population census sizes or varying portions of dimorphic flowering plants, are characterised. This is important, because high population dynamics, as typical for European orchids (cf. [40–43]) might superimpose (NFDS-driven) shifts in the morph dominance. Therefore, we report and discuss different time series, the oldest 18 years long starting in 2005, of *D. sambucina* populations in Lower Austria referring to the following questions:

(1) To what extent do the population census data in *D. sambucina* vary over time and how is this variation affecting flower colour morph frequencies?

(2) Are there differences in the dominance of yellow or red morphs at local scale, which might not be explained by large-scale environmental parameters, such as altitude and/or climatic conditions?

(3) Are shifts of the morph dominance observable in natural populations, and how might such shifts be explained?

(4) As it has been proposed that NFDS could only be detected when the rare morph becomes extremely rare (<10% frequency), and otherwise being as rare would lead to an extremely high risk of extinction of this morph, we ask whether such low frequencies do exist in the wild?

Finally, we conclude with which further analyses might be meaningful for a better understanding of the population and reproductive dynamics of *D. sambucina*.

## 2. Material and Methods

*Study Regions*

All data presented are from federal state Lower Austria in the northeast of Austria. Within Lower Austria, we observed ten *D. sambucina* populations from four different regions (Figure 2): mainly from the Waldviertel region (six populations) representing the Bohemian Massif (cf. [23,44,45]), but also one adjacent population from the Wachau region close to the Danube river, a single subpopulation from the Weinviertel region near Retz

(i.e., Mittelberg), and two from the Wienerwald region southwest of Vienna. The single populations from the Wachau and Weinviertel regions represent part of the Pannonian area, where the study species is quite rare (see [22,46,47]). In the mountainous regions of the Bohemian Massif and the Alps (here: Wienerwald region), *D. sambucina* is generally more frequent, but not common, as is also documented by its national Red List status "vulnerable" [48]. All sites represent nutrient-poor, extensively managed grasslands (i.e., meadows and pastures); however, the two Pannonian populations occur in drier and warmer conditions than those from the mountainous regions.

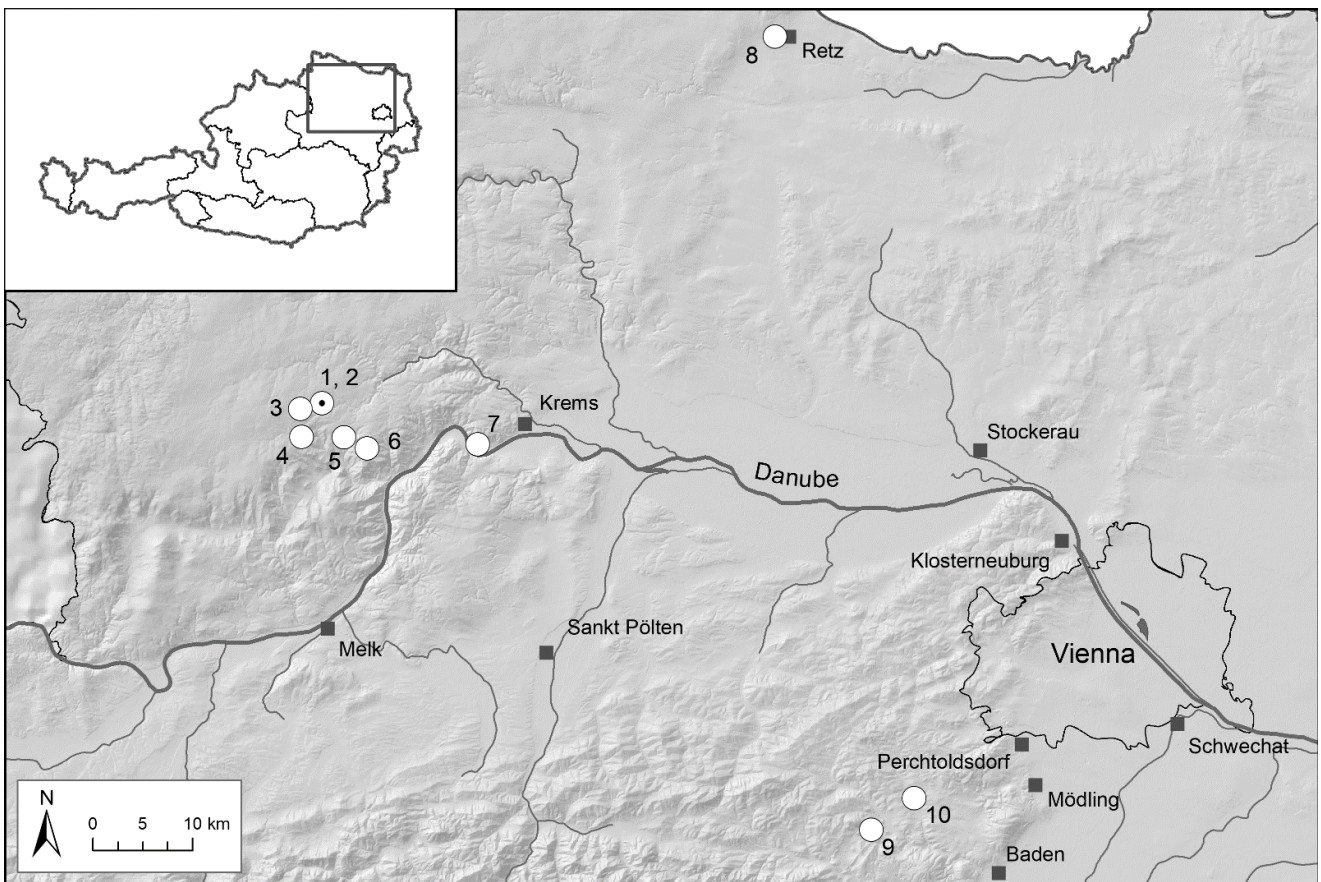

**Figure 2.** Locations (white circles) of the ten monitored *Dactylorhiza sambucina* populations in Lower Austria. The two nearby populations Voitsau I and II are marked by a dot within the white circle, as they could not be differentiated at this scale.

The longest time series started in 2005 (e.g., Waldviertel: Leopolds, Münichreith) and shorter parts of half of the time series are already published in German (i.e., from five Waldviertel populations; 2005/2006 to 2008 [23]). Overall, shorter time series cover at least nine successional study years (e.g., Wachau: Dürnstein–Höhereck; Table 1); the maximum time series from Leopolds (Waldviertel; Table 1) 18 consecutive years.

**Table 1.** Overview of *Dactylorhiza sambucina* populations with differing flower colour dominance studied, providing information about the region, the study area and years, and the different methods used.

| No. | Population | Region | Altitude (m a.s.l.) | Flower Colour Dominance | Study Area (m²) | Method Used | Study Years | Figure |
|---|---|---|---|---|---|---|---|---|
| 1 | Voitsau I | Waldviertel | 760 | yellow | 1700 | Total count of flowering plants | 2006–2022 | Figure 3, p.6 |
| 2 | Voitsau II | Waldviertel | 765 | yellow | 200 | Total count of flowering plants | 2011–2022 | Figure A3, p.14 |
| 3 | Leopolds | Waldviertel | 780 | yellow | 770 | Total count of flowering plants | 2005–2022 | Figure A1, p.13 |
| 4 | Münichreith | Waldviertel | 825 | yellow | 9000 | Total count of flowering plants | 2005–2014 | Figure A2, p.13 |
| 5 | Wernhies | Waldviertel | 670 | red | 3000 | Total count of flowering plants | 2007–2022 | Figure 4, p.7 |
| 6 | Wolfenreith | Waldviertel | 740 | red | 340 | Total count of flowering plants | 2011–2020 | Figure A4, p.14 |
| 7 | Dürnstein–Höhereck | Wachau | 285 | red | 300 | Total count of flowering plants | 2007–2015 | Figure A5, p.15 |
| 8 | Retz–Mittelberg | Weinviertel | 320 | pure yellow | 90 | Total count of all plants | 2005–2022 | Figure 6, p.9 |
| 9 | Groisbach | Wienerwald | 465 | yellow | 2800 | Total count of flowering plants | 2009–2022 | Figure A6, p.15 |
| 10 | Grub–Hocheck | Wienerwald | 435 | yellow | 2250 | Transect (225m) count of flowering plants | 2006–2022 | Figure 5, p.8 |

All study sites have repeatedly been visited in recent years by one or the other author or by both authors together. All counts were based on a single visit within each year, typically starting end of April in the Pannonian region (Weinviertel, Wachau), followed by the Wienerwald and finally the Waldviertel region with the highest altitudinal sites, i.e., typically flowering at last (Table 1). Total population census data are mainly counts of flowering plants in spatially well-defined areas (especially due to surrounding agriculture), which differ in population area size between 90 and 9000 m² (Table 1). However, one subpopulation from the Wienerwald region represents counts based on the transect technique (cf. [49]), where a defined path of 225 m length across the study site is repeated yearly and flowering plants growing five meters left and right of this way are reported, resulting in a medium area size of 2250 m² (Grub–Hocheck; Table 1). Independent of the technique used (i.e., total area count or transect count), we differentiated between yellow- and red-flowering individuals mainly, but also reported the rare salmon/pink flower colour morph, which has been interpreted as intermediate morph type [30,50]. However, as this morph is extremely rare, we did not include the single counts in our present data analyses. In the Weinviertel subpopulation, at the smallest study site (Table 1), which is characterised by yellow-flowering individuals only (cf. [47]), population census comprises both, flowering and non-flowering (i.e., vegetative) plants.

For setting general population size trends into the context of climatic conditions we investigated air temperature and precipitation data from four ZAMG meteorological stations

(https://data.hub.zamg.ac.at; access date: 5 december 2022) representing the four studied regions: i.e., from station Zwettl (Waldviertel), Krems (Wachau), Retz (Weinviertel), and Berndorf (Wienerwald). We aim at demonstrating trends of these two climatic parameters by comparing the mean monthly values of the last three decades (i.e., 1991–2000, 2001–2010, and 2011–2020).

## 3. Results

### 3.1. Study Region: Waldviertel

Remarkably, in the Waldviertel region (Figures 1 and 2), i.e., within a regional setting controlled for major climate and geology variation, we observed both, yellow- (Leopolds, Münichreith, Voitsau I + II) and red-dominated (Wernhies, Wolfenreith) *D. sambucina* populations. This observation also refers to the two most individual-rich populations there, i.e., the yellow-dominated Voitsau I population ($n_{max}$ = 1368; mean$_{17years}$ 61.8% yellow; Figure 3) and the red-dominated Wernhies population ($n_{max}$ = 519; mean$_{16years}$ 62.6% red; Figure 4). Furthermore, these two largest populations have showed no shift in the dominant flower colour morph during the time period of 16/17 consecutive study years. Moreover, these two populations—with maximal ten (1.1%) and twelve (2.4%) salmon/pink flower colour morphs within a single year—exhibited the maximum individual numbers of this extremely rare morph (cf. [30,50]).

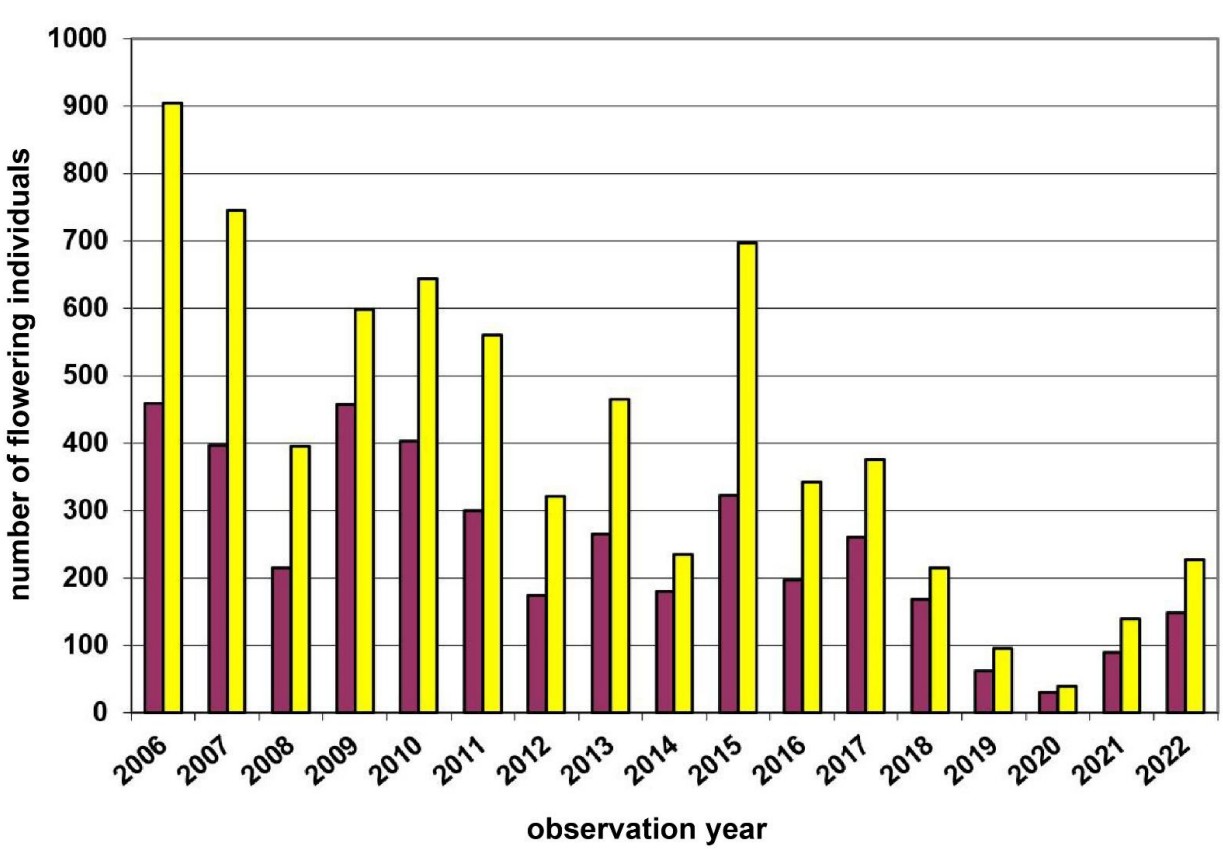

**Figure 3.** Population census (2006–2022) of the *Dactylorhiza sambucina*-population Voitsau I in the Waldviertel region. Red columns = red-flowering plants and yellow columns = yellow-flowering plants.

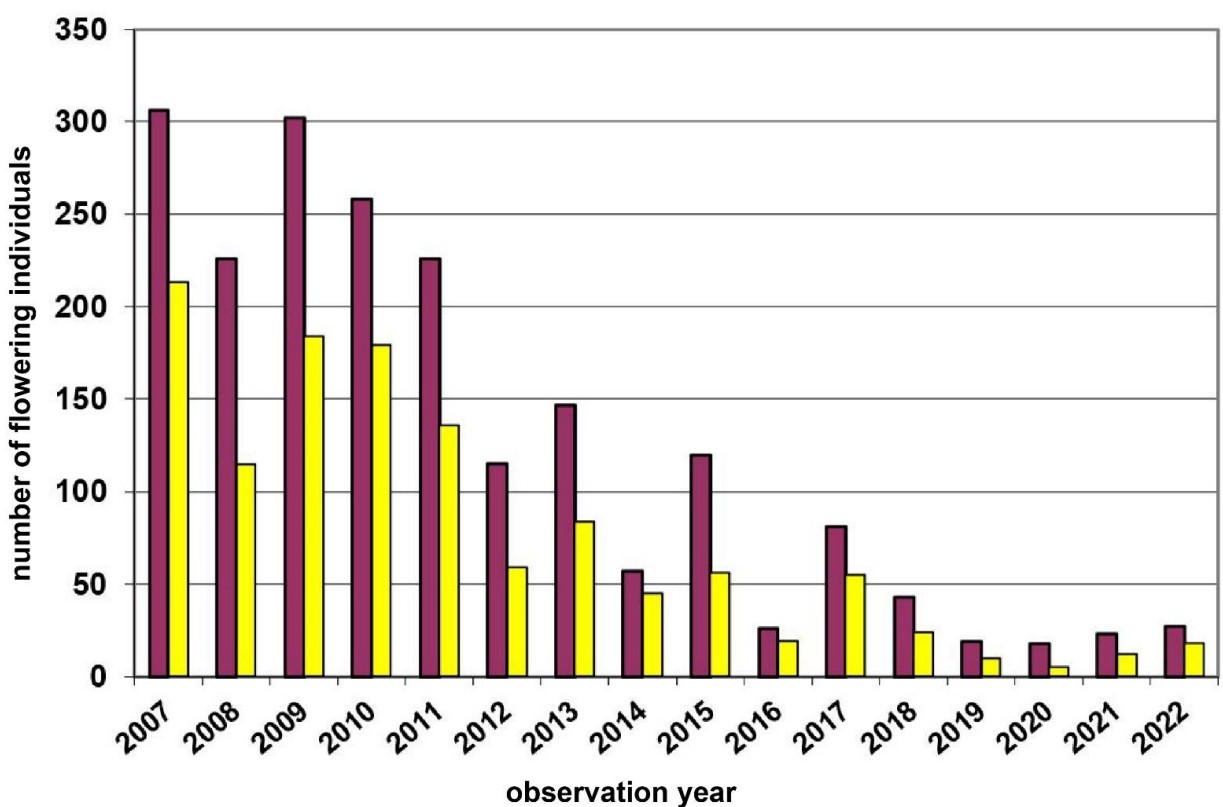

**Figure 4.** Population census (2007–2022) of the *Dactylorhiza sambucina*-population Wernhies in the Waldviertel region. Red columns = red-flowering plants and yellow columns = yellow-flowering plants.

Of the four (at first) yellow-dominated study populations in the Waldviertel region, one individual-poor population ($n_{max}$ = 70; Leopolds) has overall decreased in individual numbers, and thereby has shifted to more red-flowering plants (i.e., yellow in 2005–2010, 2013, and 2016–2017 versus red in 2012, 2014–2015, and 2018–2022; Figure A1). Another medium-sized population ($n_{max}$ = 260; Münichreith) was characterised by one shift (2008; Figure A2) towards red-dominance in a year in which the Leopolds population was still dominated by yellow-flowering plants. In contrast, the other two nearby (sub)populations, representing the most individual-rich population ($n_{max}$ = 1368; Voitsau I; Figure 3) and a medium-sized population ($n_{max}$ = 425; Voitsau II; Figure A3) whose individual numbers have declined towards almost zero between 2019–2021, have remained yellow-dominated during 17 and 12 study years, respectively. However, these two (sub)populations are spatially so close, that they are likely interconnected by gene flow. The two red-dominated populations (Wernhies, Figure 4; Wolfenreith, Figure A4) in the Waldviertel region did not show a shift in the colour morph dominance, although these two populations are very different in their population census size ($n_{max}$ = 519 versus $n_{max}$ = 19).

Independent from the flower colour morph frequencies, the population census data showed slightly varying, but negative trends: while the yellow-dominated Münichreith and the red-dominated Wernhies population have clearly decreased in total population size, the other four populations are more indifferent in their population trends. Interestingly, regarding possible explanations of the observed declines, probably two different processes are responsible: in Münichreith the numbers of flowering individuals have dramatically dropped, because the site has regularly been grazed already early in spring (i.e., during *D. sambucina* blooming) by sheep and goats, while in the Wernhies population irregular to abandoned mowing has intensified competition by co-occurring plants, including woody plants displacing *D. sambucina*.

### 3.2. Study Region: Wachau

At the southern edge of the Waldviertel region towards the Danube River, *D. sambucina* populations occur along the margin of the Pannonian biogeographical region. The small population at Höhereck near Dürnstein studied from this region, is constantly dominated by red-flowering individuals ($n_{max}$ = 48; mean$_{9years}$ 74.6% red; Figure A5). This observation is especially remarkable as there is a tendency that the red-flowering morph might dominate at higher altitudes and latitudes [35], which might mean in colder and/or more humid conditions; for more information, see [19].

### 3.3. Study Region: Wienerwald

At one study site in the Wienerwald region (Groisbach), we previously investigated the reproductive biology of *D. sambucina* during various years [51,52], and performed a pollen tracking experiment [37]. However, from one spatially defined subpopulation at this study site (patch I), we also have annual census data of yellow- and red-flowering plants since 2009 (Figure A6). During these 14 years of monitoring, counts have fluctuated between 4 (2016) and 55 (2014) flowering plants. However, despite these more than ten times fluctuations, this rather individual-poor subpopulation has always been dominated by yellow-flowering individuals (mean: 62.0%; range: 50.0–76.5%).

The second study site in the Wienerwald (Grub–Hocheck, Figure 5), where we performed transect counts from 2006 onwards, represents again a rather individual-poor subpopulation (patch III); i.e., composed of eight (2012) to 145 (2022) flowering plants within single years. This subpopulation is also mostly yellow-flowering (mean: 54.7%), but we observed a shift to more red-flowering individuals in four (i.e., 2008, 2009, 2015, and 2017) out of 15 study years. Again, these shifts are not always in the same years as those observed in the respective Waldviertel populations (see above).

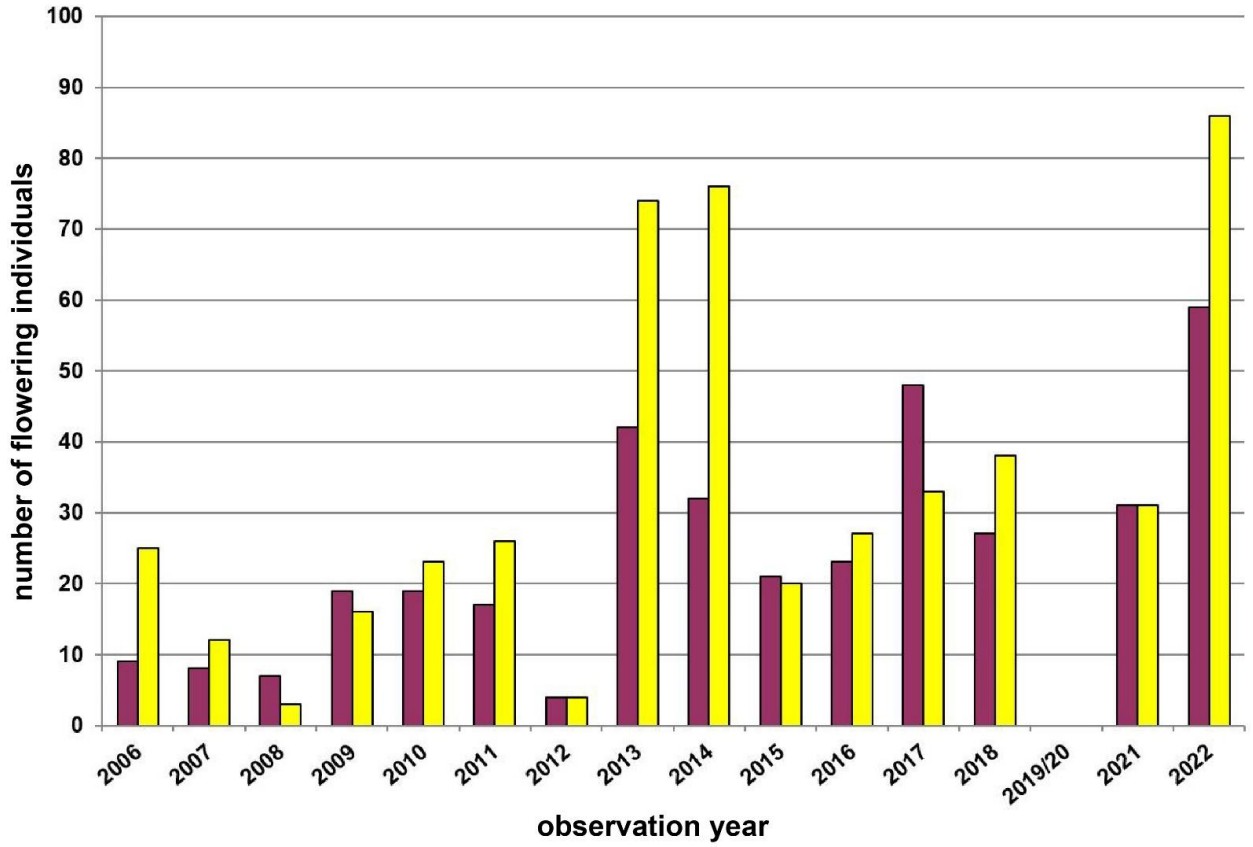

**Figure 5.** Population census (2006–2022) of the *Dactylorhiza sambucina*-population Hocheck near Grub (transect) in the Wienerwald region. This population was not visited by us in the years 2019/2020. Red columns = red-flowering plants and yellow columns = yellow-flowering plants.

### 3.4. Study Region: Weinviertel

Within the Weinviertel region, at Mittelberg near Retz, we investigated one subpopulation (patch M-VIIIa), where we counted both flowering and non-flowering individuals (Figure 6). Typically, non-flowering plants dominated (64.1–95.8%), which might indicate juvenile plants still too young to flower and/or adult plants not flowering in given years. However, in four years (i.e., 2005, 2011, 2014, and 2020) the number of flowering plants was considerably higher (59.5–74.4%), while the total census was rather constant (i.e., 37-125 individuals) over 15 consecutive study years (2008–2022).

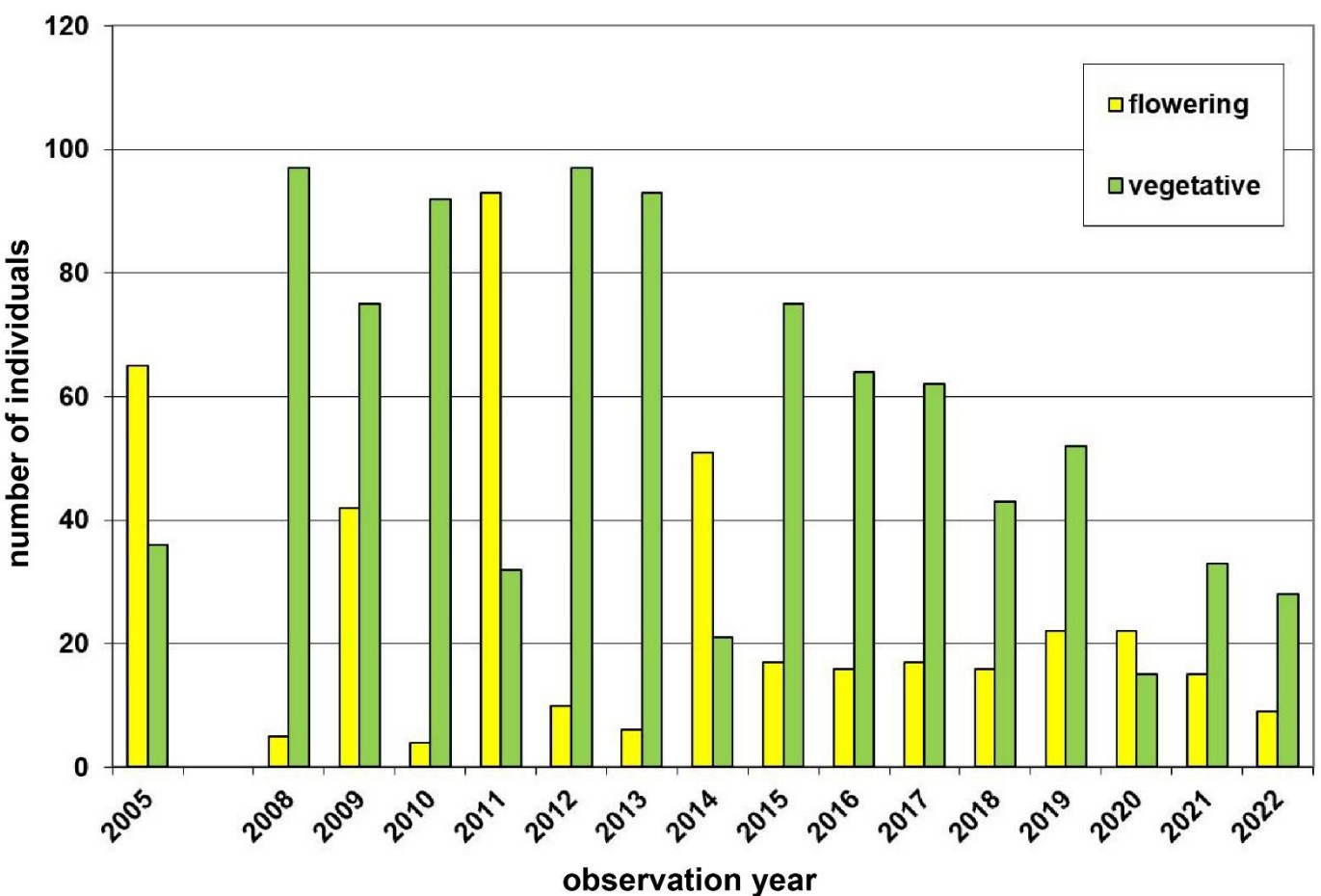

**Figure 6.** Population census (2005–2022) of the *Dactylorhiza sambucina*-population Mittelberg near Retz (patch M-VIIIa) in the Weinviertel region. This population was not visited by us in the years 2006/2007. Yellow columns = flowering plants and green columns = non-flowering, vegetative plants.

## 4. Discussion

### 4.1. Temporal Population Census Size Variation in D. sambucina

Fluctuations in flowering individuals' counts are tremendous: the factor between the lowest and the highest counts varies between four times (Wolfenreith, Figure A4) and four hundred times (Voitsau II, Figure A3). While years with the lowest counts aggregate in certain years, i.e., 2012 (Münichreith, Grub–Hocheck), 2016 (Wolfenreith, Groisbach), and 2020 (Voitsau I, Wernhies, Leopolds), and even across regions, i.e., Waldviertel and Wienerwald, the maximum numbers of populational counts were randomly distributed across years (i.e., 2006, 2007 [twice observed], 2008, 2011, 2013, 2014, 2015, and 2022). However, most of these maximum counts were more at the beginning of the respective monitoring period. Overall population size trends, therefore, although characterized by strong fluctuations, were often negative in our study populations. The only obvious exception of this trend was observed in the Wienerwald subpopulation at Hocheck near

Grub. However, this population is apparently on a way of recovery after unfavourable interventions in the 1980–90er years [53]. Overall, beside this specific situation at Hocheck, *D. sambucina* populations in the non-Alpine areas of Austria, such as in the study regions of Wald- and Weinviertel and the lower altitude Wienerwald particularly seem to suffer from (changing) climatic conditions. To demonstrate this, we investigated air temperature and precipitation data from meteorological stations nearby. Across all four study regions these climatic parameters showed dramatically decreasing precipitation—most pronounced during the last decade (comparing the last 30 years)—combined with increasing temperature specifically during March and April, the two spring months during which *D. sambucina* needs to come into leaf and blooming (see Figures A7–A10). In addition to this general trend of increasingly dry-warmer conditions during spring over the last 30 years, especially during the last decade the consecutive years of 2019–2020 were characterized by the lowest overall precipitation, likely explaining the absolute lowest population census counts in 2020 in most populations, while these populations were slightly recovered in 2022 (e.g., Voitsau I, Wernhies, Leopolds; Figures 3 and 4, Figure A1).

High population fluctuations have repeatedly been reported in European orchid species (cf. [40–43]), although most studies failed to find the ultimate causes for this behaviour, also specifically in *D. sambucina* [54]. Weather conditions are important factors (cf. [55–57]), but their influence might strongly differ depending on the timing within the season, and depending on the orchid species considered. For *D. sambucina*, as very early flowering orchid, one additional negative weather factor—beside the above discussed temperature and precipitation—repeatedly reported, is frost during budding [7,8,19,25]. Frost could stop the growing process of affected plants leading to strongly reduced rates of flowering plants within a given year. This factor was also observed in the present study, negatively influencing budding plant individuals especially in the study year 2017. However, we could not observe this factor affecting flower colour morphs in *D. sambucina* differentially; thus, shifting the flower colour morph dominance in one or the other direction during our study.

### 4.2. Flowering Dynamics in D. sambucina over Years

As the documentation of non-flowering, vegetative plant individuals remains more difficult and error-prone in the field, such data sets are much more limited. Here, we present one 16 year data set for a monochromic subpopulation from a hill, Mittelberg (325 m a.s.l.), near Retz (Weinviertel; Figure 6). While this population was rather constant in number of individuals with only 3.4 times difference between the highest (2011) and the lowest (2020, 2022) counts, the percentage of flowering individuals varied much more: at first, the proportion of flowering individuals was higher than the vegetative ones in only four years (i.e., 25%; Figure 6). Second, the observed minimum proportion of flowering plants was around 4%, while the maximum proportion was 74.4%. Moreover, the latter both extremes were observed in two subsequent years (2010 and 2011). Such tremendous differences between consecutive years might be explained by resource limitation, allowing mass flowering under good environmental conditions in one year, but forcing individual plants to recreate subsequently; thereby, skipping cost-intensive flowering and fruiting for one or more years (cf. [19,58]). However, this large variation in flowering ratios of *D. sambucina* populations, provides further difficulties, when estimating flower colour frequencies, because vegetative individuals cannot reliably be assigned to one or the other flower colour. Furthermore, it remains unknown, if both flower colour morphs behave equal in terms of propensity to flower.

Beside this erratic flowering behaviour, vegetative dormancy of adult plant individuals [59] might further create difficulties for interpreting counts of (flowering) plants as population census. This phenomenon, i.e., the below-ground survival of an adult plant individual for one to a few years is wide-spread in European terrestrial orchids. It was also reported for *D. sambucina*, with a maximum number of two [60] up to eight [19] consecutive years. Of course, this dormancy behaviour will not only influence the proportion of

flowering individuals in a given population, but also might partly explain the recovery of populations (in 2022) after the lowest census counts during the unfavourable dry years 2019–2020 (Voitsau I, Wernhies, Leopolds; Figures 3 and 4, Figure A1).

*4.3. Variation and Shifts in Flower Colour Morph Frequencies in D. sambucina*

Both yellow- and red-dominance has occurred within our Lower Austrian study region, thus indicating that large-scale environmental parameters, such as altitude and/or climatic conditions (cf. [61]), could not perfectly explain the respective flower colour dominance (see also Jersáková et al. [19]). While the mentioned population from the Mittelberg near Retz and other populations from this Weinviertel region are yellow-flowering only (cf. [47]), we studied varying yellow- and red-dominance in populations across the Waldviertel and the Wienerwald. Typically, the dominant flower colour occurs at a frequency of about two-thirds: e.g., on average 61.4% yellow-flowering in Voitsau I (Figure 3; 17 years) or 63.6% red-flowering in Wernhies (Figure 4; 16 years). Even at Münichreith, where one switch towards red-dominance was observed in 2008 (Figure A2), the averaged frequency of the yellow-flowering morph was 62.4% over the other nine study years.

As more constant population sizes might also result in more constant flower colour frequencies (cf. [23]), we also investigated this aspect. However, the most extreme variation in the appearance of flowering individuals was observed at Voitsau II ($n_{max}$ = 425 versus $n_{min}$ = 1). In this population, when excluding the two years with just one or two flowering individuals, the relation between the two colour morphs has remained very constant, with yellow-flowering individuals dominating between 73.2% in 2016 and 83.3% in 2020.

Hypothesising NFDS as significant mechanism maintaining the intraspecific polymorphism in *D. sambucina* [27], we might expect differences in the pollinator fauna in different regions of Europe playing a role or we might assume that populations observed are in different stages "oscillating" around an expected equilibrium ratio (i.e., currently red-dominated populations should become more yellow-flowering in subsequent years, and of course, also the other way round). Regarding the first assumption, evidence is clear that *D. sambucina* is very early flowering and, therefore, naïve bumblebee queens, which have repeatedly been observed as pollinators [8–10,19], are their effective pollinators. Moreover, all over Europe common species such as those from the *Bombus terrestris* complex, *B. lapidarius*, or *B. sylvarum*, are the most important pollinators (Sweden [8]; Austria [22]; France [27]; Germany [36]; Czech Republic [32]).

The second assumption seems to be refuted by the relative regional constancy of one or the other flower colour morph within different regions of Europe: while there is a dominance of yellow-flowering plants in southern France [27] and Italy [29,30], in the Czech Republic red-flowering individuals dominate [32]. In addition, by the constancy of contrary morph ratios presented in this study, at least in individual-rich populations even within the same region (i.e., Voitsau, Wernhies in the Waldviertel region), large-scale (environmental) parameters, such as geology and climate, as significant factors roughly determining the morph frequencies, are also implausible. Thus, we would not expect to find long-term constant morph frequencies around two-thirds yellow-dominated and two-thirds red-dominated within the same region (e.g., Voitsau and Wernhies are only 4.2 km air-line apart).

Documented changes of the dominant colour morph from one year to the next were observed in relatively small populations (Figure 5, Figure A1), indicating evidence for stochastic factors effecting colour morph fluctuations. The phenomenon of stochastic processes being more effective and/or having more dramatic consequences in small populations is well known from other biological processes, e.g., the higher extinction risk of populations with smaller individual numbers (and/or area size) [62–64].

Moreover, it has been proposed that NFDS could only be confirmed when the rare morph becomes extremely rare (<10% frequency [32,61]), but otherwise being that rare would lead to an extremely high risk of extinction of this morph, we asked whether such low frequencies indeed exist in the wild? Our mean morph ratios and even the ranges

across our study populations demonstrated mainly frequencies, where the infrequent morph still reached around 20–30%. The only exception was one very individual-poor population (Wolfenreith, $n_{max}$ = 19; Figure A4), where the yellow-flowering morph never reached more than 10.5%. However, within this population we could not observe a single yellow-flowering plant of *D. sambucina* during three of the last four study years. Therefore, it seems as if the extinction risk of the rare morph was higher than its reproductive success. However, we need to continue to monitor this population to exclude recovery of the rare yellow-flowering morph.

## 5. Conclusions

Regarding the results of this study, we postulate a population size of c. 500 flowering individuals where stochastic processes may not overlap population dynamics of *D. sambucina* (see also [23]). This applies particularly for the two largest study populations in the Waldviertel region (i.e., Wernhies: $n_{max}$ = 519; Voitsau: $n_{max}$ = 1364), which demonstrated no flower colour switch, and rather constant flower colour frequencies across almost two decades.

Although our data show clear evidence for mainly local processes influencing flower colour morph frequencies of *D. sambucina*, the actual relevant scale, i.e., populations or even subunits of populations, remains elusive. Rather, at a local scale, there are small-scale differences regarding colour morph frequencies. Although we are not able to verify this in detail based on our present data, we assume that even within populations there are micro-habitats with patches that may result in different colour morph dominances (pers. obs.). Therefore, we conclude here, that these intrapopulation, possibly micro-habitat related differences should be taken more into account in future studies.

Finally, the question remains which investigation periods are necessary to get valid results regarding pollinator induced selection processes and generally regarding population dynamics of *D. sambucina*. Our up to 18 years long investigation time series might still not be sufficient for a long-term population evaluation (cf. [40,43]). Respective studies rather should be based on the individual age of *D. sambucina*, which is 30 years or more [60,65–67]. This potential high age of *D. sambucina* individuals should be considered when population changes are interpreted, because actual relevant factors cannot directly be observed in the next year. Therefore, it is not surprising, that the actual flower dimorphic model organism *Linanthus parryae* (cf. [68]) is annual.

**Author Contributions:** Conceptualization, M.K. (Matthias Kropf); investigation, M.K. (Matthias Kropf) and M.K. (Monika Kriechbaum); data curation, M.K. (Matthias Kropf); writing—original draft preparation, M.K. (Matthias Kropf); writing—review and editing, M.K. (Monika Kriechbaum); visualization, M.K. (Matthias Kropf). All authors have read and agreed to the published version of the manuscript.

**Funding:** This research received no external funding.

**Institutional Review Board Statement:** Not applicable.

**Data Availability Statement:** Data is contained within the article and the Appendix A.

**Acknowledgments:** We thank Josef Pennerstorfer (BOKU Vienna) for preparing Figure 2, and the climate data in the Appendix A for us. We also thank the three Reviewers for their suggestions, which helped improve our manuscript.

**Conflicts of Interest:** The authors declare no conflict of interest.

## Appendix A

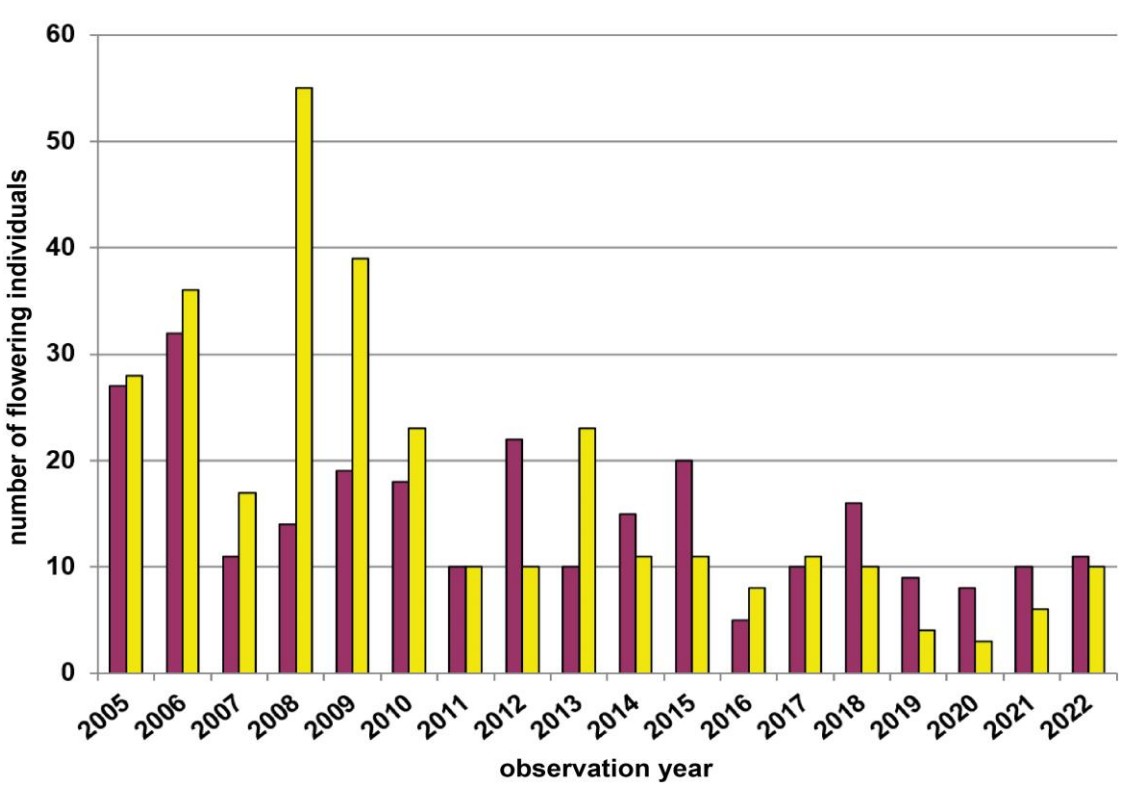

**Figure A1.** Population census (2005–2022) of the *Dactylorhiza sambucina*-population Leopolds in the Waldviertel region. Red columns = red-flowering plants and yellow columns = yellow-flowering plants.

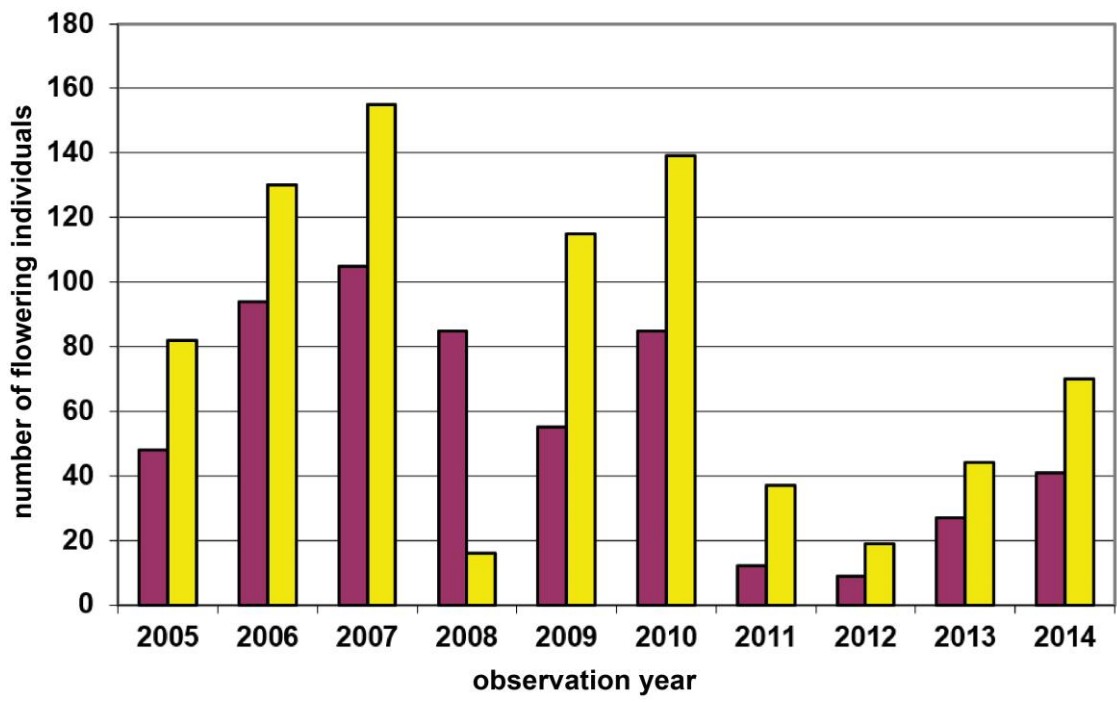

**Figure A2.** Population census (2005–2014) of the *Dactylorhiza sambucina*-population Münichreith in the Waldviertel region. Red columns = red-flowering plants and yellow columns = yellow-flowering plants.

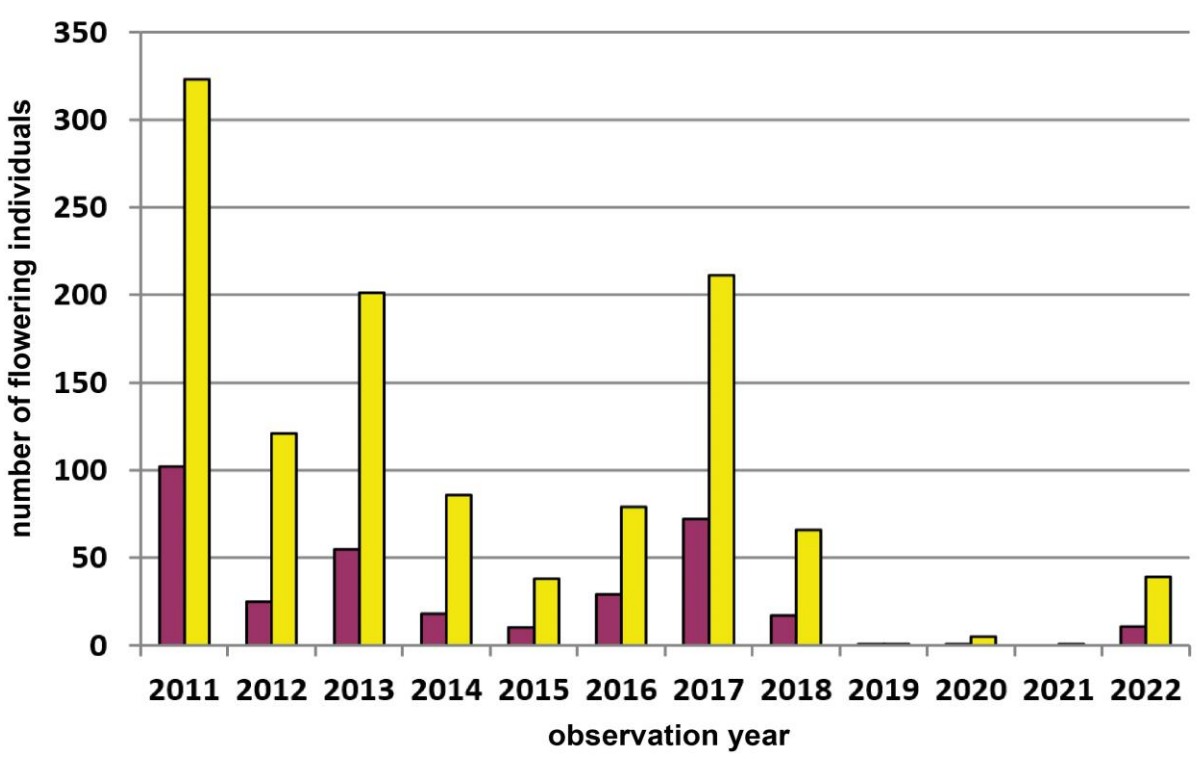

**Figure A3.** Population census (2011–2022) of the *Dactylorhiza sambucina*-population Voitsau II in the Waldviertel region. Red columns = red-flowering plants and yellow columns = yellow-flowering plants.

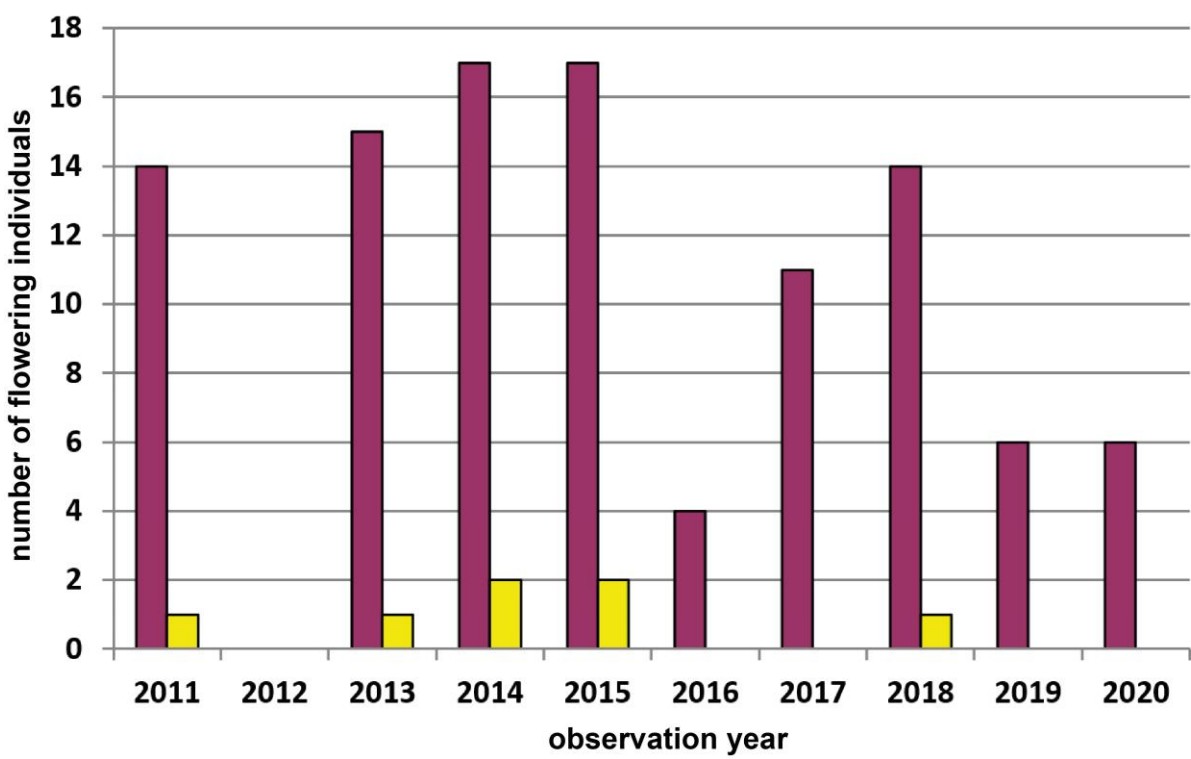

**Figure A4.** Population census (2011–2020) of the *Dactylorhiza sambucina*-population Wolfenreith in the Waldviertel region. This population was not visited by us in the year 2012. Red columns = red-flowering plants and yellow columns = yellow-flowering plants.

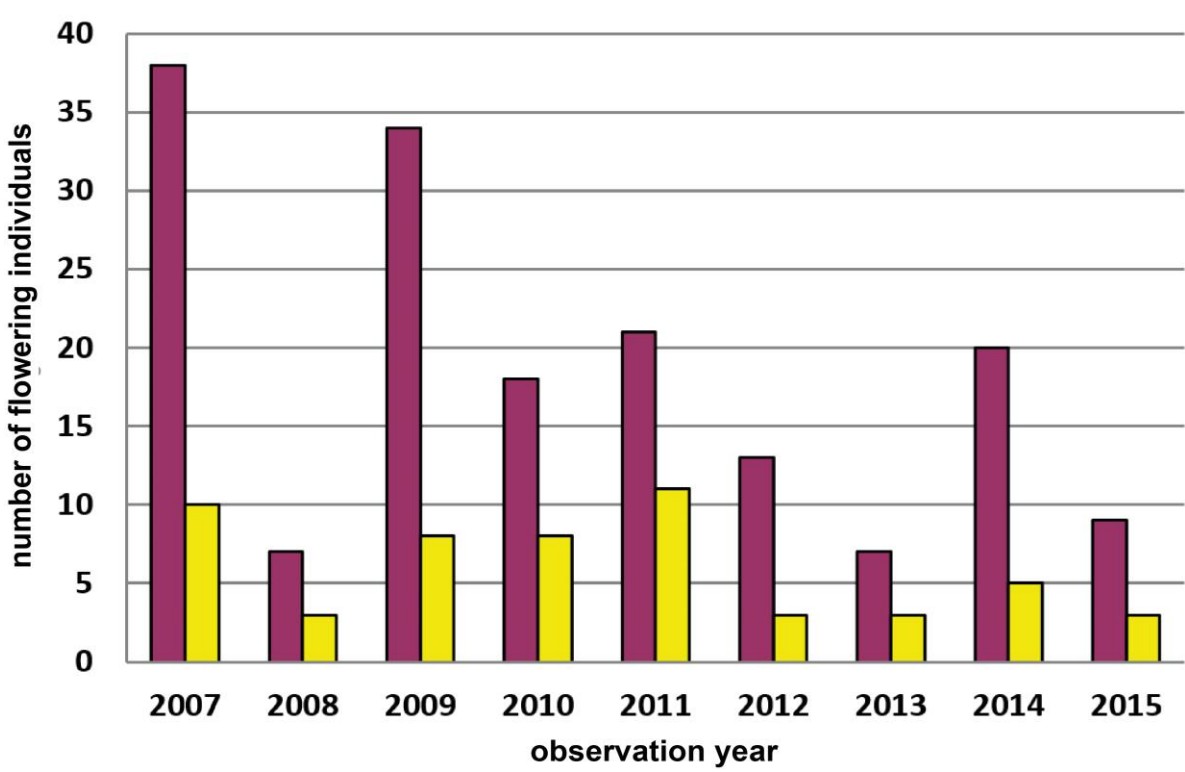

**Figure A5.** Population census (2007–2015) of *Dactylorhiza sambucina*-population Höhereck near Dürnstein in the Wachau region. Red columns = red-flowering plants and yellow columns = yellow-flowering plants.

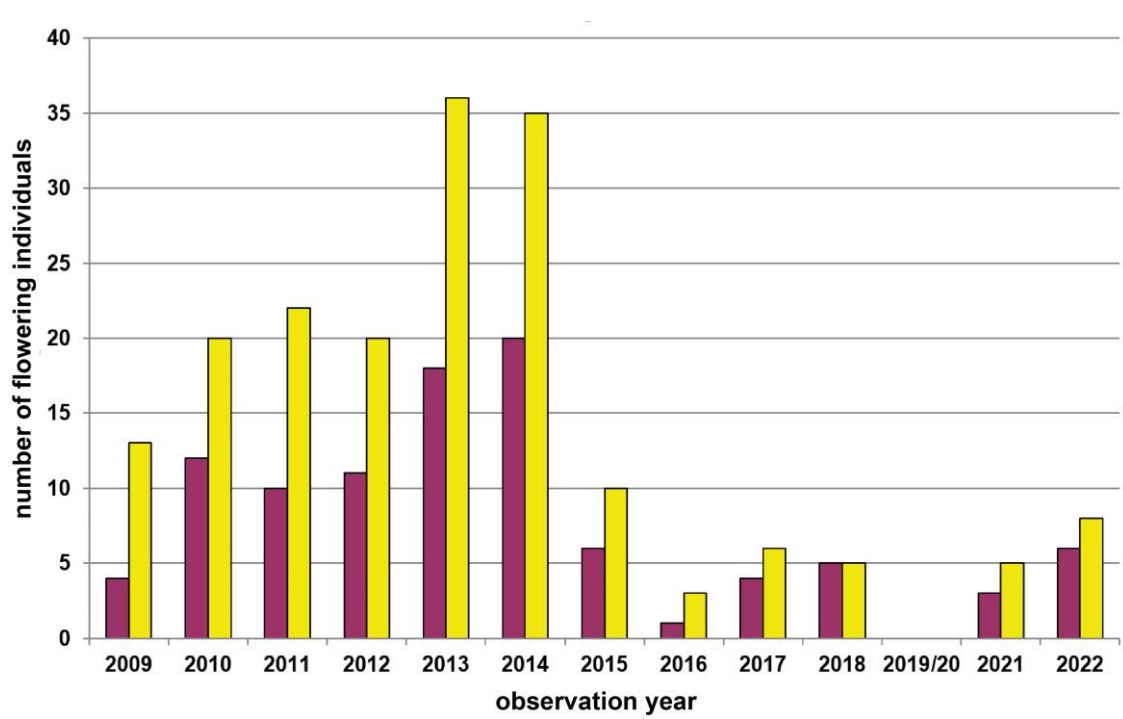

**Figure A6.** Population census (2009–2022) of the *Dactylorhiza sambucina*-population Groisbach (patch I) in the Wienerwald region. This population was not visited by us in the years 2019/2020. Red columns = red-flowering plants and yellow columns = yellow-flowering plants.

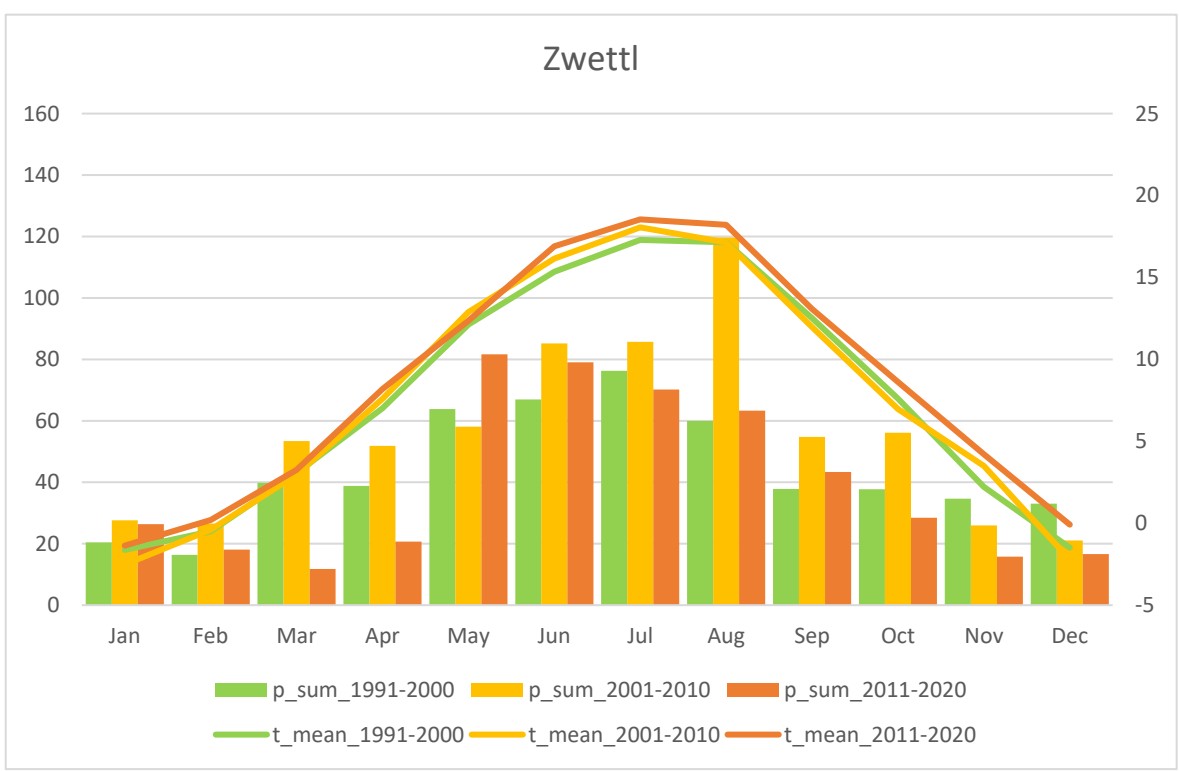

**Figure A7.** Mean monthly precipitation sum (p_sum; columns) and air temperature (t_mean; lines) for the last three recent decades, i.e. 1991–2000, 2001–2010, and 2011–2020. The "Zwettl" station is representative for the Waldviertel region (Source: ZAMG - https://data.hub.zamg.ac.at).

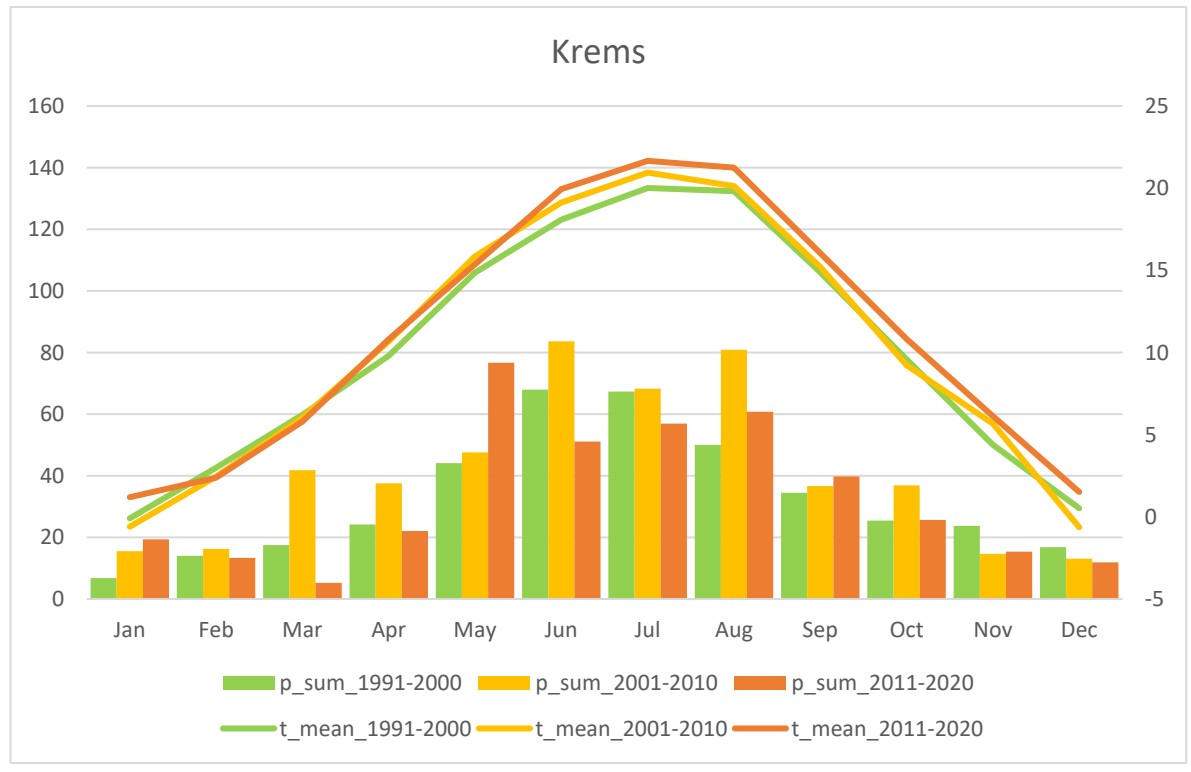

**Figure A8.** Mean monthly precipitation sum (p_sum; columns) and air temperature (t_mean; lines) for the last three recent decades, i.e. 1991–2000, 2001–2010, and 2011–2020. The "Krems" station is representative for the Wachau region (Source: ZAMG - https://data.hub.zamg.ac.at).

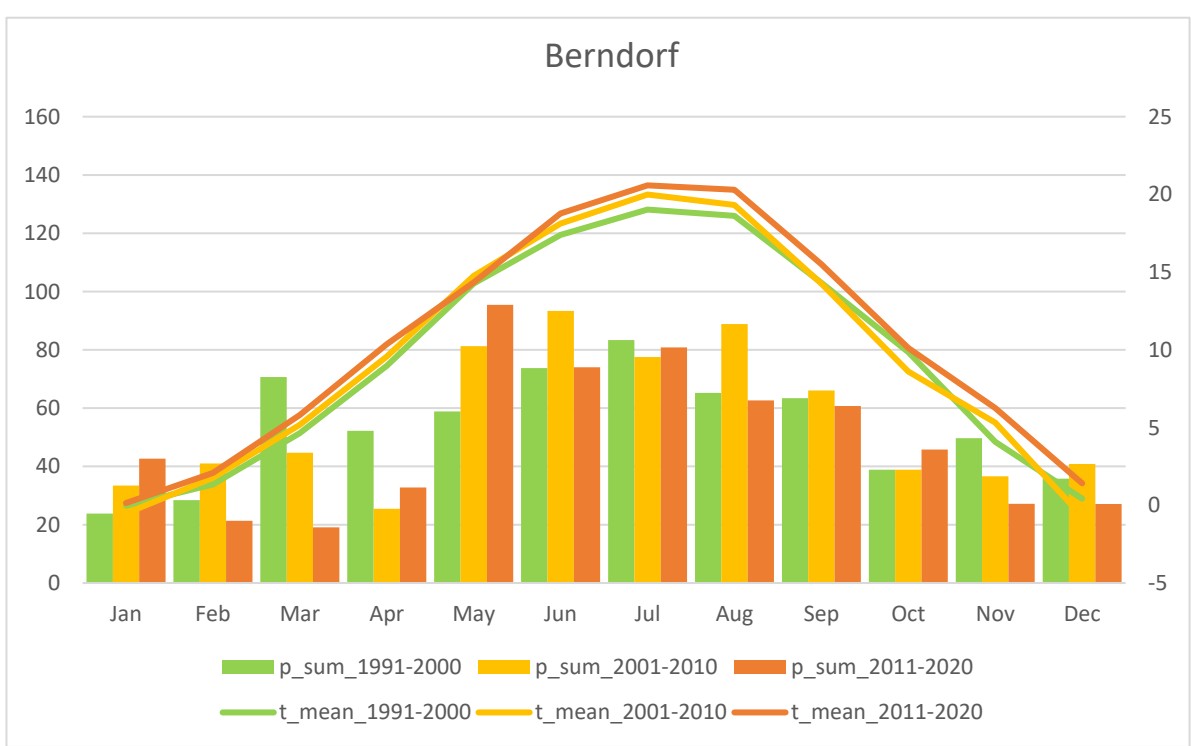

**Figure A9.** Mean monthly precipitation sum (p_sum; columns) and air temperature (t_mean; lines) for the last three recent decades, i.e. 1991–2000, 2001–2010, and 2011–2020. The "Berndorf" station is representative for the Wienerwald region (source: ZAMG - https://data.hub.zamg.ac.at).

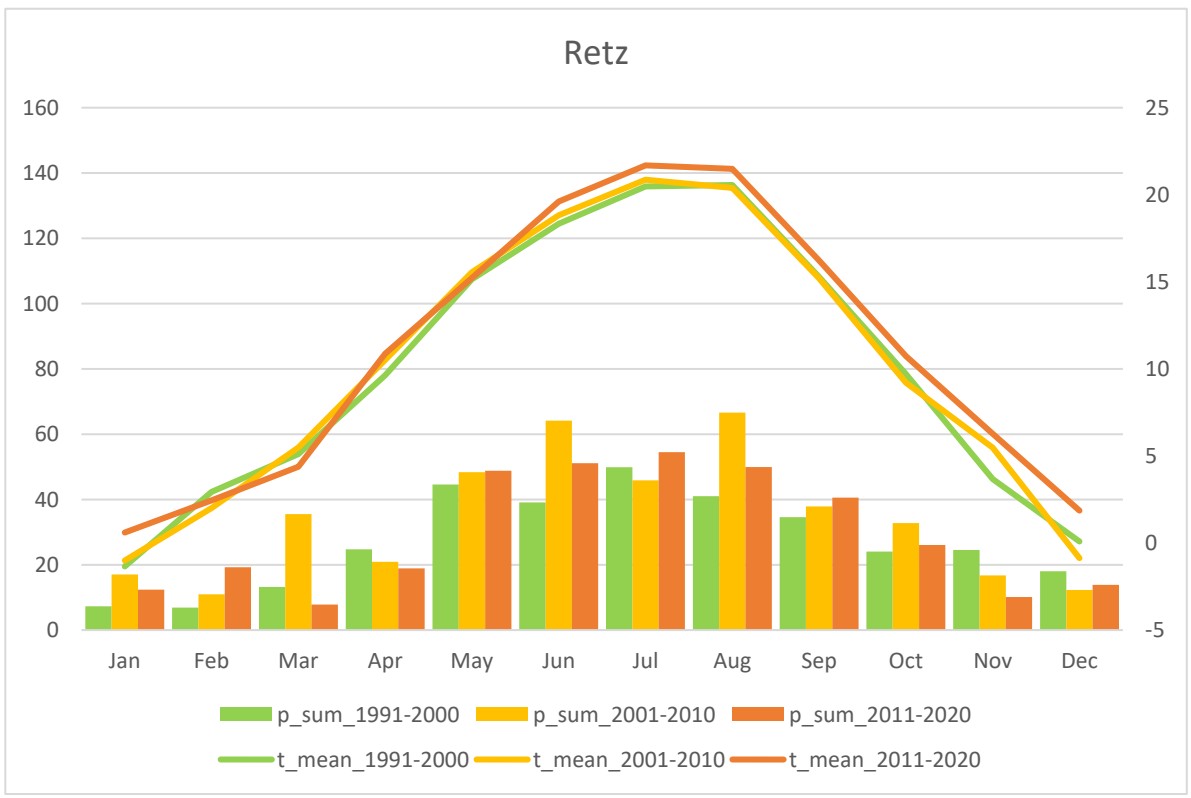

**Figure A10.** Mean monthly precipitation sum (p_sum; columns) and air temperature (t_mean; lines) for the last three recent decades, i.e. 1991–2000, 2001–2010, and 2011–2020. The Retz station is representative for the Weinviertel region (source: ZAMG - https://data.hub.zamg.ac.at).

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
