# Peer review of "Monitoring of Dactylorhiza sambucina (L.) Soó (Orchidaceae)—Variation in Flowering, Flower Colour Morph Frequencies, and Erratic Population Census Trends"

_diversity, doi:10.3390/d15020179_

Round 1

Reviewer 1 Report

The paper addresses an interesting topic about variation in flowering, flower colour morph frequencies among populations of of Dactylorhiza sambucina in the northeast of Austria. The paper is logically organized, well conducted using correct methodology and well written, although there are some points that deserve some improvement.

First, Materials and Methods: the authors need to clarify the procedure for total population census data. Does the reported value relate to the count of a single day per year in each population? Thus, it would seem given the lack of standard error or standard deviation in the figures.  Or are the values ​​an average of multiple observations? It is important to clarify this aspect as the observation period can influence the number of flowering plants.

Another aspect to bring into the discussion is the relationships with biotic factors such as pollinators, mycorrhizae and other sympatric plants (the magnet species effect). They are all factors closely related to reproductive success and therefore to the dynamics of orchid populations

Author Response

Reviewer 1

The paper addresses an interesting topic about variation in flowering, flower colour morph frequencies among populations of of Dactylorhiza sambucina in the northeast of Austria. The paper is logically organized, well conducted using correct methodology and well written, although there are some points that deserve some improvement.

Reply: Thanks for this benevolent statement.

First, Materials and Methods: the authors need to clarify the procedure for total population census data. Does the reported value relate to the count of a single day per year in each population? Thus, it would seem given the lack of standard error or standard deviation in the figures.  Or are the values an average of multiple observations? It is important to clarify this aspect as the observation period can influence the number of flowering plants.

Reply: We clarified these methodical aspects and moreover, added further information as also suggested by another Reviewer to the Material and Methods chapter, including a new Table, for instance, summarizing the area sizes (sqm) studied at each site.

Another aspect to bring into the discussion is the relationships with biotic factors such as pollinators, mycorrhizae and other sympatric plants (the magnet species effect). They are all factors closely related to reproductive success and therefore to the dynamics of orchid populations

Reply: The Reviewer is basically right, that a number of different abiotic and biotic factors are also influencing the reproductive success of orchid populations. However, as we didn’t analyze the reproductive success here, but only discuss expectations of flower colour morph frequencies (fluctuations) in D. sambucina populations in the context with hypotheses on the maintenance of this polymorphism, we provided respective information basically only at the beginning of our manuscript, i.e. in the Introduction and the Study species chapters (and do not further discuss this later, because we did not collect any data in this respect; i.e. concrete reproductive success data). However, as also another Reviewer would like to see more details and references to studies already performed with D. sambucina, we comprehensively rephrased the Introduction and for instance added also different aspects on pollinators, pollination strategy, and the influence of co-flowering (magnet) plant species. But, we did not also discuss mycorrhizae, as this topic is too far beyond our investigations.

Reviewer 2 Report

The manuscript offers interesting insights on the reproduction dynamics of the reward less Dactylorhiza sambucina (Orchidaceae) to what concerns the pollinator-mediated negative frequency dependent selection and its role in maintaining the flower colour polymorphism.

The introduction needs to be implemented by providing a complete state of art for the study aims.

Other minor corrections are then reported.

Overall, the manuscript deserves publication following a revision.

L12: please, remove the bold effect on the first word.

L19: complete the sentence, ‘play an important role’ for what…?

L22: I am not sure about this form, please I would suggest checking the form or to clarify it: ‘prone to dominance switches’.

L37: please, review the sentence: the bumblebees are cheated in which way? Explain the pollination mechanism. The sentence does not provide explanation about.

L41: please clarify the sentence as is not clear how cross pollination is favoured. Also, the sentence seems to be interrupted.

L56: please integrate the following accepted study to support contrasting evidence that the rarer phenotype, obtained manipulating UV reflectance signals, did not impact the visiting pollinator rate comparing to dominant phenotype in case the food mimicking orchids were nearby their model plants (Scaccabarozzi et al., 2023). So, the effect might be mimic-model distance dependent.

Title: Mimicking orchids lure bees from afar with exaggerated ultraviolet signals, Accepted in Journal: Ecology and Evolution

Authors

Daniela Scaccabarozzi, Klaus Lunau, Lorenzo Guzzetti, Salvatore Cozzolino, Adrian G. Dyer, Nicola Tommasi, Paolo Biella, Andrea Galimberti, Massimo Labra, Ilaria Bruni,  Giorgio Pattarini, Mark Brundrett, Monica Gagliano

DOI: 10.1002/ece3.9759

L69: may the morph diversity depend on model plants morph variation and/or abundance? This aspect should be integrated referring to the appropriate literature.

Some suggested literature on effect of model plants abundance on orchid reproductive success:

Smithson, Ann, Nicolas Juillet, Mark R. Macnair, and Luc DB Gigord. "Do rewardless orchids show a positive relationship between phenotypic diversity and reproductive success?." Ecology 88, no. 2 (2007): 434-442.

Scaccabarozzi, Daniela, Lorenzo Guzzetti, Ryan D. Phillips, Lynne Milne, Nicola Tommasi, Salvatore Cozzolino, and Kingsley W. Dixon. "Ecological factors driving pollination success in an orchid that mimics a range of Fabaceae." Botanical Journal of the Linnean Society 194, no. 2 (2020): 253-269.

L85: could you include any aim related to the potential model plants’ effect? Of course in case you have any available data.

L190: please, correct the subtitle as it sounds unclear as it is.

L220 please, correct into ‘Flowering variations (or fluctuations) in D. sambucina over years’.

L309: please, remove italic.

Author Response

Reviewer 2

The manuscript offers interesting insights on the reproduction dynamics of the reward less Dactylorhiza sambucina (Orchidaceae) to what concerns the pollinator-mediated negative frequency dependent selection and its role in maintaining the flower colour polymorphism.

Reply: Okay, thanks.

The introduction needs to be implemented by providing a complete state of art for the study aims.

Reply: We added further concepts (e.g. the exact mode of the deceptive pollination strategy of our study species) and related concepts, like floral mimicry and the magnet-species effect in the Introduction chapter. This also includes further references. However, we basically still stick to the state of the art specifically in relation to our species-specific study system.

Other minor corrections are then reported.

Overall, the manuscript deserves publication following a revision.

Reply: Thank you for this statement!

L12: please, remove the bold effect on the first word.

Reply: Done.

L19: complete the sentence, ‘play an important role’ for what…?

Reply: We added “…influencing reproduction”.

L22: I am not sure about this form, please I would suggest checking the form or to clarify it: ‘prone to dominance switches’.

Reply: Rephrased: “…prone to changes of flower colour dominance”

L37: please, review the sentence: the bumblebees are cheated in which way? Explain the pollination mechanism. The sentence does not provide explanation about.

Reply: We already indicated that our study species doesn’t provide a nectar reward; however, we now added the information that the species morphologically indeed is characterized by having a flower spur, which is empty. Furthermore, we added a respective sentence indicating that these characteristics correspond to the generalized food deceptive strategy of this species; and added general and specific respective references.

L41: please clarify the sentence as is not clear how cross pollination is favoured. Also, the sentence seems to be interrupted.

Reply: It is indicated in this sentence, that “pollinator switches between morphs” are promoted and therefore favour cross-pollination. There is no interruption within this sentence, but we made two out of one sentence.

L56: please integrate the following accepted study to support contrasting evidence that the rarer phenotype, obtained manipulating UV reflectance signals, did not impact the visiting pollinator rate comparing to dominant phenotype in case the food mimicking orchids were nearby their model plants (Scaccabarozzi et al., 2023). So, the effect might be mimic-model distance dependent.

Title: Mimicking orchids lure bees from afar with exaggerated ultraviolet signals, Accepted in Journal: Ecology and Evolution

Authors

Daniela Scaccabarozzi, Klaus Lunau, Lorenzo Guzzetti, Salvatore Cozzolino, Adrian G. Dyer, Nicola Tommasi, Paolo Biella, Andrea Galimberti, Massimo Labra, Ilaria Bruni,  Giorgio Pattarini, Mark Brundrett, Monica Gagliano

DOI: 10.1002/ece3.9759

L69: may the morph diversity depend on model plants morph variation and/or abundance? This aspect should be integrated referring to the appropriate literature.

Some suggested literature on effect of model plants abundance on orchid reproductive success:

Smithson, Ann, Nicolas Juillet, Mark R. Macnair, and Luc DB Gigord. "Do rewardless orchids show a positive relationship between phenotypic diversity and reproductive success?." Ecology 88, no. 2 (2007): 434-442.

Scaccabarozzi, Daniela, Lorenzo Guzzetti, Ryan D. Phillips, Lynne Milne, Nicola Tommasi, Salvatore Cozzolino, and Kingsley W. Dixon. "Ecological factors driving pollination success in an orchid that mimics a range of Fabaceae." Botanical Journal of the Linnean Society 194, no. 2 (2020): 253-269.

Reply: We suggest the Editor to check for the rules of the MDPI journals, which basically discourage from recommending own publications to be added as citations.

Our point is, that the suggested references (2x Scaccabrozzi et al.; see above), although very interesting and well-performed investigations, treated a food-deceptive orchid genus (i.e. Diuris, here: D. brumalis and D. magnifica), which is obviously mimicking co-occurring Fabaceae models; moreover, not being flower colour dimorphic. The situation in D. sambucina is therefore different in multiple respects, although the species is also food-deceptive, because our study species represents a so-called “generalized food deceptive system” (i.e. without a specific model plant). Already Nilsson (1980) failed to identify model plant species for D. sambucina (although he listed some potential and at his sites co-occurring plants), as did Kropf (1996), although (or therefore…) not deeply discussed in subsequent publications (Kropf & Renner 2005, 2008). However, we now made this point (i.e. the generalized food deceptive strategy) more clear in the Introduction chapter, stressing the aspect of non-model-specific food deception, including basic references. Moreover, we also added further references relating to the magnet species effect, as well as different experimental studies, including the by the Reviewer suggested work of Smithson et al. (2007).

L85: could you include any aim related to the potential model plants’ effect? Of course in case you have any available data.

Reply: There is no model plant, as has been shown by different studies (now cited)!

L190: please, correct the subtitle as it sounds unclear as it is.

Reply: It is unclear for us, why this subheading “sounds unclear”? However, we rephrased by moving the terminal “over time” phrase as “temporal” at the beginning of the subheading. Better?

L220 please, correct into ‘Flowering variations (or fluctuations) in D. sambucina over years’.

Reply: We found an alternative formulation, which hopefully reads better:
Flowering dynamics of D. sambucina over years

L309: please, remove italic.

Reply: Why? It is possible style to strengthen a statement. The Journal editor might decide whether the style of the diversity Journal allows this stylistic device.

Reviewer 3 Report

The article addresses issues of great conservation and theoretical importance, based on long-term population monitoring. The results of the research presented in the article are relevant for many researchers working on population dynamics and conservation issues in the Orchidaceae. Despite the strengths of the article, there are a number of points that need to be improved.

1. Keywords need to be changed and expanded, as most of them now repeat the terms in the title of the article. 

2. The abstract needs to be improved to make the aims and methods of the study clear to the reader and to briefly present the main results. The current text is more of an annotation than a true abstract. 

3. I would suggest starting the introduction with a broader theoretical background to the subject and then moving on to a consideration of research on Dactylorhiza sambucina. 

4. I recommend that the authors not only mention the different frequencies of morphs in different European countries, but also refer to them and give at least a few examples (lines 39-41). 

5. In the introduction, as throughout the text, the authors should avoid vague references to literature sources without saying what controversies these sources deal with (line 42; [...] but see [21].). 

6. The authors discuss "negative frequency-dependent selection" but do not explain this theory in detail. In the introduction, I think the most important aspects of this theory should be discussed, as it is one of the highlights of the paper (lines 43-44). 

7. I would like to draw the authors' attention to the need to be very precise in the use of terms when referring to pollination and cross-pollination (line 52). A mother plant may be referred to as a pollen acceptor, a paternal individual may be referred to as a pollen donor, and when this is not taken into account they may be referred to generically as parental individuals.

8. What did the authors mean by "Even after correcting" (line 58)? I suggest that the authors should be very precise in their sentences to leave no room for broad interpretations, and that the information should be presented in such a way that it is not necessary to search for cited sources for each statement. 

9. The Materials and Methods section needs a major reorganisation and additions. I suggest that information on the habitats of each population should be presented in tabular form, indicating the type of habitat, the area covered, the altitude and other relevant environmental parameters. 

10. On the map, I propose to number the study sites according to the future numbering in the table. 

11. I recommend that a clear hierarchical system of terms is established and used consistently throughout the text. The authors should define what constitutes a site, a locality and a population. I also suggest that the use of quotation marks for geographical names should be avoided. After all, place names are not used figuratively, they are not symbolic names and they are not quotations. 

12. I missed a detailed description of the field study in the methodology. It is described that the transect method was used to survey plants in a 10 m wide belt. In this case, were only the individuals of the transect included in the number of individuals, or were others outside the belt included? How were the plants surveyed in other cases? This is important information, not only for those carrying out similar surveys, but also to evaluate the accuracy and reliability of this survey. 

13. There is no methodology for statistical treatment of the data. It is therefore not clear what the authors call e.g. nmax? In my opinion, it is necessary to outline the methods of data processing in a very clear and coherent manner. 

14. Why not try to make a statistical comparison of the composition of populations according to the colour of the flowers of individuals. There are statistical methods available that allow comparisons even between different sample sizes and allow us to determine whether differences between populations and/or survey years are significant or insignificant.

15. I would suggest including a figure in the methodology with photographs showing the variation in flower colour and how the variation is referred to in the article.  

16. In all the graphs in the text of the article and in the supplementary material, there is no indication of what is on the y-axis. All graphs need to be updated. 

17. The results section needs to be cleaned up by removing the discussion elements (e.g. lines 161-163). 

18. What does [2x] mean? (line 195).

19. Have the authors not tried to express the absolute number of individuals by maturity phase (Figure 5) in percentages? Perhaps such an expression would be more informative. By the way, the methodology should describe how the maturity groups of individuals were assessed, over what area and how the study was conducted in the wild.

20. I would suggest that the conclusions should be more concise and that the discussion should be moved away from the conclusions. The conclusions should not refer to publications, especially without explaining the context (line 296). 

21. The scientific names and authors of all organisms mentioned in the text should be given with precision, and ambiguous naming should be avoided (line 269; Bombus terrestris/lucorum). 

The article has great potential, but it needs a major revision.

Author Response

Reviewer 3

The article addresses issues of great conservation and theoretical importance, based on long-term population monitoring. The results of the research presented in the article are relevant for many researchers working on population dynamics and conservation issues in the Orchidaceae. Despite the strengths of the article, there are a number of points that need to be improved.

  1. Keywords need to be changed and expanded, as most of them now repeat the terms in the title of the article.

Reply: Thank you for this correction hint; we added two keywords not repeating the title.

  1. The abstract needs to be improved to make the aims and methods of the study clear to the reader and to briefly present the main results. The current text is more of an annotation than a true abstract.

Reply: We made several changes in our Abstract; also, according to suggestions made by other Reviewers. However, we are not sure what the Reviewer means by “annotations” here? Basically, our Abstract follows the usual structure, like in the main text body.

  1. I would suggest starting the introduction with a broader theoretical background to the subject and then moving on to a consideration of research on Dactylorhiza sambucina.

Reply: As suggested by the Reviewer we added a more general Introduction-starting point on pollination biology of orchids and then moved into our specific orchid study species.

  1. I recommend that the authors not only mention the different frequencies of morphs in different European countries, but also refer to them and give at least a few examples (lines 39-41).

Reply: We added information about the main flower colour biases in the different countries.

  1. In the introduction, as throughout the text, the authors should avoid vague references to literature sources without saying what controversies these sources deal with (line 42; [...] but see [21].).

Reply: We now indicate the contradicting results of the respective study, by adding “… which could not demonstrate by pollen tracking that pollinators alternate randomly between colour morphs.”

  1. The authors discuss "negative frequency-dependent selection" but do not explain this theory in detail. In the introduction, I think the most important aspects of this theory should be discussed, as it is one of the highlights of the paper (lines 43-44).

Reply: We cannot follow this criticism, as in the first version of the manuscript, the idea of NDFS with the specific consequences for the maintenance of the flower colour dimorphism in D. sambucina is described in a small paragraph running over seven lines (i.e. 43-48). In the present version the lines 61-67.

Naturally, NDFS is also relevant in other biological situations. However, it is not clear, why this theory should not be introduced species-specifically, at this point?!

  1. I would like to draw the authors' attention to the need to be very precise in the use of terms when referring to pollination and cross-pollination (line 52). A mother plant may be referred to as a pollen acceptor, a paternal individual may be referred to as a pollen donor, and when this is not taken into account they may be referred to generically as parental individuals.

Reply: We are precise. We wrote “mother plants” – so, we meant “pollen acceptor”.
However, we do not think, that the term “mother plant” is imprecise in this respect, as the Reviewer also stated this, respectively; however, we now used both terms to make it hopefully more clear.

  1. What did the authors mean by "Even after correcting" (line 58)? I suggest that the authors should be very precise in their sentences to leave no room for broad interpretations, and that the information should be presented in such a way that it is not necessary to search for cited sources for each statement.

Reply: Within the statistics used in the mentioned publication, when testing for significant differences between random expectations and observed values, the different morph frequencies of the population studied were used as respective products for all four possible crosses, representing expectations. A Chi2-test was used to test differences between these expected values and the observed values by pollen tracking. However, we do not think, that this statistical procedure needs to be described here; but, nevertheless, we added “statistically” and mentioned the “Chi2-test”, to make more clear, that within the cited publication as part of a statistical testing different morph frequencies were considered (= “corrected for”).

  1. The Materials and Methods section needs a major reorganisation and additions. I suggest that information on the habitats of each population should be presented in tabular form, indicating the type of habitat, the area covered, the altitude and other relevant environmental parameters.

Reply: We added a respective Table 1, and appreciated this suggestion. However, we described the habitat types not in this table, but in the respective text body.

  1. On the map, I propose to number the study sites according to the future numbering in the table.

Reply: We added a numbering in the new Table 1, and we also used this numbers on the revised map.

  1. I recommend that a clear hierarchical system of terms is established and used consistently throughout the text. The authors should define what constitutes a site, a locality and a population. I also suggest that the use of quotation marks for geographical names should be avoided. After all, place names are not used figuratively, they are not symbolic names and they are not quotations.

Reply: We tried to use a clearer, consistent (and hierarchically structured) terminology, and checked the text throughout, respectively. Geographically, we used the defined regions (as also indicated in the new Table 1), as the larger units. Within these regions we have our study sites (we also used the term sites for the environmental description of these sites) and on these sites the study populations (the term we preferentially used), which sometimes might be further subdivided as spatially nearby, but still differentiable units (subpopulations) – for instance: the site Voitsau is differentiated into population Voitsau I and Voitsau II (see Table 1).

We no longer use location or locality; with the only exception of the legend of the map, where location basically means the geographical position.

We cancelled the annotations for the population names as suggested by the Reviewer.

  1. I missed a detailed description of the field study in the methodology. It is described that the transect method was used to survey plants in a 10 m wide belt. In this case, were only the individuals of the transect included in the number of individuals, or were others outside the belt included? How were the plants surveyed in other cases? This is important information, not only for those carrying out similar surveys, but also to evaluate the accuracy and reliability of this survey.

Reply: As we now provide the study area sizes (sqm) in the new Table 1, it should be more clear, that these population areas we completely monitored for all occurring (flowering) D. sambucina plants. Yes, of course, is a transect restricted to the by us provided dimension of the transect. Otherwise, when including plant individuals outside the transect, it would have been a total area count, too.

  1. There is no methodology for statistical treatment of the data. It is therefore not clear what the authors call e.g. nmax? In my opinion, it is necessary to outline the methods of data processing in a very clear and coherent manner.

Reply: Nmax is of course the maximum count within a given year throughout the respective monitoring time period. We don’t think, that this is unclear.

  1. Why not try to make a statistical comparison of the composition of populations according to the colour of the flowers of individuals. There are statistical methods available that allow comparisons even between different sample sizes and allow us to determine whether differences between populations and/or survey years are significant or insignificant.

Reply: It is not clear which concrete statistical testing we might use at this point; or even, what should be tested statistically? The “composition” is always just yellow- and red-flowering in counts of individual plants, which could of course be expressed as percentages. However, as (larger) yellow-dominated populations, like Voitsau I (Figure 3) remained yellow-dominated within each of the 17 consecutive study years, and (larger) red-dominated populations, like Wernhies (Figure 4) remained red-dominated within each of the 16 consecutive study years – representing one major outcome of our monitoring – it seems unclear, what might be tested here, statistically, to make this unambiguous observation more rigorous?

  1. I would suggest including a figure in the methodology with photographs showing the variation in flower colour and how the variation is referred to in the article.

Reply: Thanks for this suggestion. We included a new Figure 1 into our manuscript demonstrating the yellow- and red-flowering D. sambucina plants within a Waldviertel population – the region, where most of our study population are located. This picture is now also used as background for the Graphical Abstract.

  1. In all the graphs in the text of the article and in the supplementary material, there is no indication of what is on the y-axis. All graphs need to be updated.

Reply: We updated all respective Figures by adding labels for both axes.

  1. The results section needs to be cleaned up by removing the discussion elements (e.g. lines 161-163).

Reply: The mentioned lines give a pure description of at first the situation in the population Leopolds, indicating its population size, its decline over time, and the shift regarding the dominance of the flower colour over the same time, while the second mentioned population Münichreith showed only one shift in one year, and we further indicated, that this year was another year compared to Leopolds. This basically descriptive providing observational results. It would have been a part of a Discussion, if we would have added, for instance, the hypothesis, that these observations indicate that under varying climatic conditions (in different years) shifts towards red-dominance are possible and that therefore other factors might have caused these shifts…

This is the respective paragraph to check our opinion (= pure Results):

“Of the four (at first) yellow-dominated study populations in the Waldviertel region, one individual-poor population (nmax = 70; Leopolds) has overall decreased in individual numbers and thereby has shifted to more red-flowering plants (i.e., yellow in 2005–2010, 2013, 2016–2017 versus red in 2012, 2014–2015, 2018–2022; Figure A1). Another medium-sized population (nmax = 260; Münichreith) was characterised by one shift (2008; Figure A2) towards red-dominance in a year in which the Leopolds population was still dominated by yellow-flowering plants.”

  1. What does [2x] mean? (line 195).

Reply: “2x” means two times. It is a usual abbreviation.

  1. Have the authors not tried to express the absolute number of individuals by maturity phase (Figure 5) in percentages? Perhaps such an expression would be more informative. By the way, the methodology should describe how the maturity groups of individuals were assessed, over what area and how the study was conducted in the wild.

Reply: Regarding the differentiation between vegetative and flowering individual plants, we first have to note, that we could not differentiate between age of individuals. So, individual plants were not marked individually in the field, but within a given area all individuals are simply surveyed, counted and differentiated between flowering and not-flowering (= vegetative). And this is clearly stated in the M&M chapter.

Of course, as for the two flower colour morphs, all counts could also be expressed as percentages (for each year), but we would like to present the data similarly across all Figures.

With respect to the question which area was covered, we now provide this information in the new Table 1 (as thankfully suggested by the Reviewer). And from this table it becomes obvious, that the time-consuming count of flowering and vegetative plants was only possible on a smaller area size (here: 90 sqm).

  1. I would suggest that the conclusions should be more concise and that the discussion should be moved away from the conclusions. The conclusions should not refer to publications, especially without explaining the context (line 296).

Reply: The mentioned citation (original line 296; new line 335) is our own previous work, which is also mentioned at the beginning of the manuscript (new line 123), because for five Waldviertel populations shorter time series (max. five years) were already published in German. And here, we pick up parts of the (previous) Conclusions and therefore included this citation.

The Conclusions cover three aspects: i) the role of population sizes, ii) the role of intrapopulation, micro-habitat variability, and iii) the role of the needed length of the time series – and these aspects are not discussed here, but we suggest that future investigations might draw more attention to these aspects – the cited publications just confirm our conclusions.

  1. The scientific names and authors of all organisms mentioned in the text should be given with precision, and ambiguous naming should be avoided (line 269; Bombus terrestris/lucorum).

Reply: Why? All zoologists know that the differentiation of taxa within the Bombus terrestris complex is not always straightforward; therefore, it is quite usual, that these taxa are not all the time differentiated, precisely (especially by botanists). However, we know use the term Bombus terrestris complex, although we know that this includes potentially more than the two most common species, i.e. B. terrestris and B. lucorum mentioned before!

The article has great potential, but it needs a major revision.

Reply: We hope that our comprehensively revised manuscript version, now meets the expectations of the Reviewers and the Editors.

Round 2

Reviewer 2 Report

I am happy with the manuscript publication.

Author Response

Reviewer 2:

I am happy with the manuscript publication.

Reply: We thank the Reviewer for her/his final positive statement.

Reviewer 3 Report

I would like to thank the authors for their explanations and answers to the questions raised in the previous review. The current version of the paper has been significantly improved and I have no substantive comments on its content. A few technical and editorial comments remain. 

1. In a previous review, I wrote that the current abstract is more like an abstract (an essay about what is being studied but without presenting the results; a kind of promotional text). The current version of the abstract is also possible, but usually consists of a statement of the problem, the aim of the study, a very brief methodology, the main results and the main conclusion. This comment is in response to a question raised in the authors' reply to a comment made in the previous review.

2. The units of area in the text should be given as usual in the scientific literature (m2 but not sqm). Applies to Table 1 and the following text. 

3. I would like to object to the authors on the clarity of the use of 2x. This is usually used to refer to populations or individuals with a 2x basic chromosome set, and sometimes it is used to refer to double magnification, so it is not really clear in this context. Especially as the correction requires a minimum of effort, with 2x in place ('twice studied'). 

4. The text of the article needs careful technical editing (spacing of references, standardised use of dashes and hyphens, quotation marks, etc.). 

All other comments made in the previous review have been explained or taken into account. Even if I have a slightly different opinion on some of the points under discussion, the authors of the article have sufficient reasons to defend their opinion. For this reason, I believe that the article can be accepted. 

Author Response

Reviewer 3:

I would like to thank the authors for their explanations and answers to the questions raised in the previous review. The current version of the paper has been significantly improved and I have no substantive comments on its content. A few technical and editorial comments remain.

Reply: We are pleased, that we have been able to clarify different points, and thank the Reviewer again for her/his valuable suggestions which helped improving our manuscript!

  1. In a previous review, I wrote that the current abstract is more like an abstract (an essay about what is being studied but without presenting the results; a kind of promotional text). The current version of the abstract is also possible, but usually consists of a statement of the problem, the aim of the study, a very brief methodology, the main results and the main conclusion. This comment is in response to a question raised in the authors' reply to a comment made in the previous review.

Reply: We rephrased the central part of the Abstract again partly, for introducing more precise information (and numbers), explaining what we did. The subsequent (after the rephrased sentence) three sentences provide information about results (i.e. number of populations showing changes in the dominant flower morph, synchronism of these switches in different years, and answering the question which populations are more prone for those changes).

  1. The units of area in the text should be given as usual in the scientific literature (m2 but not sqm). Applies to Table 1 and the following text. 

Reply: We changed sqm to m2 throughout the manuscript as suggested by the Reviewer.

  1. I would like to object to the authors on the clarity of the use of 2x. This is usually used to refer to populations or individuals with a 2x basic chromosome set, and sometimes it is used to refer to double magnification, so it is not really clear in this context. Especially as the correction requires a minimum of effort, with 2x in place ('twice studied'). 

Reply: We changed 2x to ‚twice observed‘ following the suggestion by the Reviewer.

  1. The text of the article needs careful technical editing (spacing of references, standardised use of dashes and hyphens, quotation marks, etc.). 

Reply: We (again) went through the whole manuscript for technical editing. Please, find respective corrections in the track changes mode.

All other comments made in the previous review have been explained or taken into account. Even if I have a slightly different opinion on some of the points under discussion, the authors of the article have sufficient reasons to defend their opinion. For this reason, I believe that the article can be accepted. 

Reply: We would like to thank the Reviewer for this fair final judgement, although we don’t think that both opinions are as far away from each other, as this statement might imply.